# DeiSAM: Segment Anything with Deictic Prompting

**Hikaru Shindo**[1][*] **Manuel Brack**[1,2] **Gopika Sudhakaran**[1,3]
**Devendra Singh Dhami**[4] **Patrick Schramowski**[1,2,3,5] **Kristian Kersting**[1,2,3]
[1]Technical University of Darmstadt [2]German Research Center for AI (DFKI)
[3]Hessian Center for AI (hessian.AI) [4]Eindhoven University of Technology
[5]Center for European Research in Trusted Artificial Intelligence (CERTAIN)

## Abstract

Large-scale, pre-trained neural networks have demonstrated strong capabilities in various tasks, including zero-shot image segmentation. To identify concrete objects in complex scenes, humans instinctively rely on *deictic* descriptions in natural language, *i.e.*, referring to something depending on the context, such as "The object that is on the desk and behind the cup". However, deep learning approaches cannot reliably interpret such deictic representations as they have limited reasoning capabilities, particularly in complex scenarios. Therefore, we propose DeiSAM—a combination of large pre-trained neural networks with differentiable logic reasoners—for deictic promptable segmentation. Given a complex, textual segmentation description, DeiSAM leverages Large Language Models (LLMs) to generate first-order logic rules and performs differentiable forward reasoning on generated scene graphs. Subsequently, DeiSAM segments objects by matching them to the logically inferred image regions. As part of our evaluation, we propose the Deictic Visual Genome (DeiVG) dataset, containing paired visual input and complex, deictic textual prompts. Our empirical results demonstrate that DeiSAM is a substantial improvement over purely data-driven baselines for deictic promptable segmentation.

## 1 Introduction

Recently, large-scale neural networks have substantially advanced various tasks at the intersection of vision and language. One such challenge is grounded image segmentation, wherein objects within a scene are identified through textual descriptions. For instance, Grounding Dino (Liu et al., 2023c), combined with the Segment Anything Model (SAM) (Kirillov et al., 2023), excels at this task if provided with appropriate prompts. However, a well-documented limitation of data-driven neural approaches is their lack of reasoning capabilities (Shi et al., 2023; Huang et al., 2024a). Consequently, they often fail to understand complex prompts that require high-level reasoning on relations and attributes of multiple objects, as demonstrated in Fig. 1.

In contrast, humans identify objects through structured descriptions of complex scenes referring to an object, *e.g.*, "An object that is on the boat and holding an umbrella". These descriptions are referred to as *deictic representations* and were introduced to artificial intelligence research motivated by linguistics (Agre & Chapman, 1987), and subsequently applied in reinforcement learning (Finney et al., 2002). A deictic expression refers to an object depending on the agent using it and the overall context. Although deictic representations play a central role in human comprehension of scenes, current approaches fail to interpret them faithfully due to their poor reasoning capabilities.

To remedy these issues, we propose DeiSAM, which is a combination of large pre-trained neural networks with differentiable logic reasoners for deictic promptable object detection and segmentation.

---

[*]corresponding author: hikaru.shindo@tu-darmstadt.de

38th Conference on Neural Information Processing Systems (NeurIPS 2024).

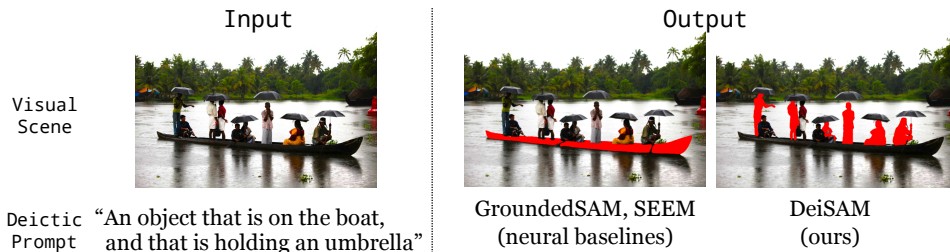

Figure 1: **DeiSAM segments objects with deictic prompting.** Shown are segmentation masks with an input textual prompt. DeiSAM (right) correctly segments the *people* on the boat holding umbrellas, whereas the neural baselines (left) incorrectly segment the *boat* instead (Best viewed in color).

The DeiSAM pipeline is highly modular and fully differentiable, sophisticatedly integrating large pre-trained networks and neuro-symbolic reasoners. Specifically, we leverage Large Language Models (LLMs) to generate logic rules for a given deictic prompt and perform differentiable forward reasoning (Shindo et al., 2023, 2024) with scene graph generators (Zellers et al., 2018). The reasoner is efficiently combined with neural networks by leveraging forward propagation on computational graphs. The result of this reasoning step is used to ground a segmentation model that reliably identifies the objects best matching the input.

In summary, we make the following contributions: 1) We propose DeiSAM[2], a modular, neuro-symbolic framework using LLMs and scene graphs for object segmentation with complex textual prompts. 2) We introduce a novel Deictic Visual Genome (DeiVG) benchmark that contains visual scenes paired with deictic representations, *i.e.*, complex textual identifications of objects in the scene. To further investigate the challenging nature of abstract prompts, we curate a new DeiRefCOCO+ benchmark. It is a deictic variant of RefCOCO+, an established reference object detection benchmark. 3) We empirically demonstrate that DeiSAM strongly outperforms neural baselines for deictic segmentation. 4) We showcase that DeiSAM can perform end-to-end training via differentiable reasoning to improve the segmentation quality adapting to complex downstream reasoning tasks.

## 2 Related Work

**Multi-modal Large Language Models.** The recent achievements of large language models (LLMs) (Brown et al., 2020) have led to the development of multi-modal models, including vision-language models (Radford et al., 2021; Alayrac et al., 2022; Li et al., 2022a; Liu et al., 2023b), which take visual and textual inputs. However, these large models' reasoning capabilities are limited (Huang et al., 2024a), often inferring wrong conclusions when confronted with complex reasoning tasks. DeiSAM addresses these issues by combining large models with (differentiable) reasoners.

Additionally, DeiSAM is related to prior work using LLMs for program generation. For example, LLMs have been applied to generate probabilistic programs (Wong et al., 2023), Answer Set Programs (Ishay et al., 2023; Yang et al., 2023), and programs for visual reasoning (Surís et al., 2023; Stanić et al., 2024). These works have demonstrated that LLMs are powerful program generators and outperform simple zero-shot reasoning. With DeiSAM we propose the usage of LLMs to generate differentiable logic programs for image segmentation and object detection.

**Scene Graph Generation.** Scene Graph Generators (SGGs) encode complex visual relations to a summary graph using the comprehensive contextual knowledge of relation encoders (Lu et al., 2016; Zellers et al., 2018; Tang et al., 2019). Recently, the focus has shifted to transformer-based SGGs that use attention to capture global context while improving visual and semantic fusion (Lin et al., 2020; Lu et al., 2021; Dong et al., 2022). Lately, attention has also been used to capture object-level relation cues using visual and geometric features (Sudhakaran et al., 2023). The modularity of DeiSAM allows for using any SGG to obtain graph representations of input visual scenes. Scene graphs are essential for segmentation models to be faithful reasoners. Without them, models may develop shortcuts, resulting in apparent answers through flawed scene understanding (Marconato et al., 2023).

**Visual Reasoning and Segmentation.** Visual Reasoning has been a fundamental problem in machine learning research, resulting in multiple benchmarks (Antol et al., 2015; Johnson et al., 2017; Yi et al.,

---

[2]Code: `https://github.com/ml-research/deictic-segment-anything`

2020) to address this topic and subsequent frameworks (Yi et al., 2018; Mao et al., 2019; Amizadeh et al., 2020; Hsu et al., 2023) that perform reasoning using symbolic programs and multi-modal transformers (Tan & Bansal, 2019). These benchmarks are primarily developed to answer queries written in natural language texts paired with visual inputs. Our proposed dataset, DeiVG, is the first to integrate complex textual prompts into the task of image segmentation with natural images. In a similar vein, to tackle visual reasoning tasks, neuro-symbolic rule learning frameworks have been proposed, where discrete rule structures are learned via backpropagation (Evans & Grefenstette, 2018; Minervini et al., 2020; Shindo et al., 2021, 2023, 2024; Zimmer et al., 2023). These works have primarily been tested on visual arithmetic tasks or synthetic environments for reasoning (Stammer et al., 2021). To this end, LASER (Huang et al., 2024b) leverages logical specifications to learn properties from videos. DeiSAM is a unique neuro-symbolic framework that addresses image segmentation in natural images and utilizes differentiable reasoning for program learning.

Semantic segmentation aims to generate objects' segmentation masks given visual input (Wang et al., 2018; Guo et al., 2018). Multiple datasets and tasks have been proposed that assess a model's reasoning ability to identify objects (Kazemzadeh et al., 2014; Yu et al., 2016). Recently, Segment Anything Model (SAM) (Kirillov et al., 2023) has been released, achieving strong results on zero-shot image segmentation tasks. Grounded SAM (Ren et al., 2024) combines Grounding DINO (Liu et al., 2023c) with SAM, allowing for objects described by textual prompts. Moreover, LISA (Lai et al., 2023) fine-tunes multi-modal LLMs to perform low-level reasoning over image segmentation. However, LISA still requires strong prior information on the type target object (*e.g.* "the *person* that is wearing green shoes") and breaks down for more abstract tasks (*cf.* Sec. 5.5). In contrast, DeiSAM encodes the reasoning process explicitly as a differentiable function, thus avoiding spurious neural networks' behavior. Consequently, DeiSAM is capable of high-level reasoning on arbitrarily abstract prompts (*e.g.* "an *object*") utilizing structured representation of scene graphs. To this end, frameworks that enhance the transformer (or attention) architecture for various segmentation tasks have been proposed (Liu et al., 2023a; Wu et al., 2024a,b). These approaches rely on transformers (or attentions) as their core reasoning pipeline. In contrast, DeiSAM explicitly encodes logical reasoning processes to guarantee accurate and faithful interpretation of abstract and complex prompts.

# 3 DeiSAM — The Deictic Segment Anything Model

DeiSAM uses first-order logic as its language, and we provide its formal definition in App. A. Let us start by outlining the DeiSAM pipeline with a brief overview of its modules, before describing essential components in more detail.

## 3.1 Overview: Deictic Segmentation

We show a schematic overview of the proposed DeiSAM workflow in Fig. 2. First, an input image is transferred into a graphical representation using a **(1) Scene Graph Generator**. Specifically, a scene graph comprises a set of triplets $(n_1, e, n_2)$, where entities $n_1$ and $n_2$ have relation $e$. For example, a *person* $(n_1)$ is *holding* $(e)$ an *umbrella* $(n_2)$. Consequently, each triplet $(n_1, e, n_2)$ in a scene graph can be interpreted as a fact, $\texttt{e(n}_1\texttt{, n}_2\texttt{)}$, where $\texttt{e}$ is a 2-ary predicate and $\texttt{n}_1$ and $\texttt{n}_2$ are constants in first-order logic. The textual deictic prompt needs to be interpreted as a structured logical expression to perform reasoning on these facts.

For this step, DeiSAM leverages **(2) Large Language Models**, which can generate logic rules for deictic descriptions, given sufficiently restrictive prompts as we demonstrate. In our example, the LLM would translate "An object that is on the boat,

```
// Program 1
cond1(X):-on(X,Y),type(Y,boat).
cond2(X):-holding(X,Y),type(Y,umbrella).
target(X):-cond1(X),cond2(X).
```
Listing 1: Rules generated by LLMs.

and that is holding an umbrella" into the rules (`Program 1`) in Listing 1. The first two rules define the conditions described in the prompt, and the last rule identifies corresponding objects. However, users often use terminology different from that of the SGG, *e.g.*, *boat* and *barge* target the same concept but will not be trivially matched. To bridge the semantic gap, we introduce a **(3) semantic unifier**. This module leverages word embeddings of labels, entities, and relations in the generated scene graphs and rules to match synonymous terms by modifying rules accordingly. The semantically unified rules are then compiled to a **(4) forward reasoner**, which computes logical entailment using forward chaining (Shindo et al., 2023). The reasoner identifies the targeted objects and their bounding

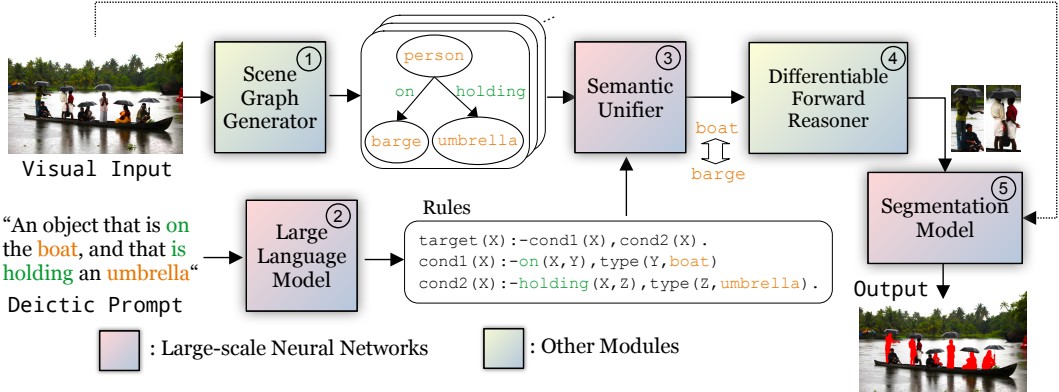

Figure 2: **DeiSAM architecture.** An image paired with a deictic prompt is given as input. We parse the image into a scene graph **(1)** and generate logic rules **(2)** corresponding to the deictic prompt using a large language model. The generated scene graph and rules are fed to the *Semantic Unifier* module **(3)**, where synonymous terms are unified. For example, `barge` in the scene graph and `boat` in the generated rules will be interpreted as the same term. Next, the forward reasoner **(4)** infers target objects specified by the textual deictic prompt. Lastly, we perform object segmentation **(5)** on extracted cropped image regions of the target objects. Since the forward reasoner is differentiable (Shindo et al., 2023), gradients can be passed through the entire pipeline (Best viewed in color).

boxes from the scene graph. Lastly, we segment the object by feeding the cropped images to a **(5) segmentation model**.

Now, let us investigate the two core modules of DeiSAM in detail: rule generation and reasoning.

## 3.2 LLMs as Logic Generators

To perform reasoning on textual prompts, we need to identify corresponding rules. We use LLMs to parse textual descriptions to logic rules using the system prompt specifying the rule format to be generated. The complete prompt is provided in App. B. DeiSAM uses a specific rule format describing object and attribute relations. For example, a fact `on(person,boat)` in a scene graph would be decomposed into multiple facts `on(X,Y)`, `type(X,person)`, and `type(Y,boat)` to account for several entities with the same attribute in the scene.

The computational and memory cost of forward reasoning is determined by the number of variables over all rules and the number of conditions. Naive formatting of rules (Shindo et al., 2024) leads to an exponential resource increase with the growing complexity of deictic prompts. Since the representations used in the forward reasoner are pre-computed and kept in memory, non-optimized approaches will quickly lead to exhaustive memory consumption (Evans & Grefenstette, 2018). In our format, however, we restrict the used variables to `X` and `Y` and only increase the number of rules with growing prompt complexity. Thus resulting in *linear* scaling of computational costs instead.

## 3.3 Reasoning with Deictic Prompting

DeiSAM performs differentiable forward reasoning as follows. We build a reasoning function $f_{reason} : \mathcal{G} \times \mathcal{R} \rightarrow \mathcal{T}$ where $\mathcal{G}$ is a set of facts representing a scene graph, $\mathcal{R}$ is a set of rules generated by an LLM, and $\mathcal{T}$ is a set of facts representing identified target objects in the scene.

**(Differentiable) Forward Reasoning.** For a visual input $x \in \mathbb{R}^2$, DeiSAM utilizes scene graph generators (Zellers et al., 2018) to obtain a logical graph representation $\mathcal{G}$, where each fact `rel(obj1,obj2)` $\in \mathcal{G}$ represents an edge in the scene graph. Each fact in a given set $\mathcal{G}$ is mapped to a confidence score using a *valuation vector* $v \in [0,1]^{|\mathcal{G}|}$. A SGG is a function $sgg : \mathbb{R}^2 \rightarrow [0,1]^{|\mathcal{G}|}$ that produces a valuation vector out of a visual input. DeiSAM builds on the neuro-symbolic message-passing reasoner (NEUMANN) (Shindo et al., 2024) to perform reasoning. For a given set of rules $\mathcal{R}$, DeiSAM constructs a *forward reasoning graph*, which is a bi-directional graph representation of a logic program. Given an initial valuation vector produced by an SGG, DeiSAM computes logical consequences in a differentiable manner by performing bi-directional message passing on

the constructed reasoning graph using soft-logic operations (*cf.* App. A.1). DeiSAM identifies target objects to be segmented using confidence scores over facts representing targets, *e.g.*, `target(obj1)`, and extracts the corresponding bounding boxes from the scene graph.

**Semantic Unifier.** DeiSAM unifies diverging semantics in the generated rules and scene graph using concept embeddings similar to neural theorem provers (Rocktäschel & Riedel, 2017). We rewrite the corresponding rules $\mathcal{R}$ of a prompt by identifying the most similar terms in the scene graph for each predicate and constant. If rule $R \in \mathcal{R}$ contains a term $x$, which does not appear in scene graph $\mathcal{G}$, we compute the most similar term as $\arg\max_{y \in \mathcal{G}} encoder(x)^\top \cdot encoder(y)$, where $encoder$ is an embedding model for texts. We apply this procedure to terms and predicates individually.

# 4    The Deictic Visual Genome

To facilitate a thorough evaluation of the novel deictic object segmentation tasks, we introduce the Deictic Visual Genome (DeiVG) dataset. Building on Visual Genome (Krishna et al., 2017), we construct pairs of deictic prompts and corresponding object annotations for real-world images, as shown in Fig. 3. Our analysis of the scene graphs in Visual Genome found the annotations to often be noisy and ambiguous, which aligns with observations from previous research (Hudson & Manning, 2019). Consequently, we substantially filtered and cleaned potential candidates to produce a sound dataset.

An object that has a handle and that is on a bench

Figure 3: An example from Deictic Visual Genome (DeiVG$_2$).

First, we restricted ourselves to 19 commonly occurring relations and ensured that prompts were unambiguous, with only one kind of target object satisfying the prompt. Specifically, DeiVG contains prompts requiring the correct identification of multiple objects, but these are guaranteed to be the same type according to Visual Genomes synset annotations. We automatically synthesize prompts from the filtered scene graphs using textual templates, *e.g.*, the relations `has(cooler,handle)` and `on(cooler,bench)` would yield a prompt "An object that has a handle and that is on a bench" targeting the cooler. Entries in the DeiVG dataset can be categorized by the number of relations they use in their object description. We introduce three subsets with 1-3 relations, which we denote as DeiVG$_1$, DeiVG$_2$, and DeiVG$_3$, respectively. Each dataset is distinct, *e.g.*, DeiVG$_2$ contains only prompts using 2 relations. For each set, we randomly select 10k samples that we make publicly available to encourage further research.

# 5    Experimental Evaluation

With the methodology of DeiSAM and our novel evaluation benchmark DeiVG established, we now provide empirical and qualitative experiments. Our results outline DeiSAM's benefits over purely neural approaches, supplemented by ablation studies of each module. Additionally, we investigate RefCOCO (Yu et al., 2016), a low-level reasoning benchmark for segmentation tasks, and demonstrate the robustness of DeiSAM for abstract prompts. Lastly, we show that DeiSAM is end-to-end trainable and can thus be leveraged to improve the performance of the neural components in the pipeline.

## 5.1    Experimental Setup

We base our experiments on the three subsets of DeiVG. As an evaluation metric, we use mean average precision (mAP) over objects. Since the object segmentation quality largely depends on the used segmentation model, we focus on assessing the object identification preceding the segmentation step. The default DeiSAM configuration for the subsequent experiments uses the ground truth scene graphs from Visual Genome (Krishna et al., 2017), `gpt-3.5-turbo`[3] as LLM for rule generation, `ada-002`[4] as embedding model for semantic unification, and SAM (Kirillov et al., 2023) for object segmentation. Additionally, we provide few-shot examples of deictic prompts and paired rules in the input context of the LLM, which improves performance (*cf.* App. E). We present detailed ablations on each component of the DeiSAM pipeline in Sec. 5.4.

---

[3]`https://openai.com/blog/introducing-chatgpt-and-whisper-apis`

[4]`https://openai.com/blog/new-and-improved-embedding-model`

Table 1: **DeiSAM handles deictic prompting.** Mean Average Precision (mAP) of DeiSAM and neural baselines on DeiVG datasets are shown. Subscript numbers indicate the complexity of prompts.

| Method | Mean Average Precision (%) ↑ | | |
|---|---|---|---|
| | DeiVG$_1$ | DeiVG$_2$ | DeiVG$_3$ |
| SEEM | 1.58 | 4.44 | 7.54 |
| OFA-SAM | 3.37 | 9.01 | 15.38 |
| GLIP-SAM | 2.32 | 0.03 | 0.00 |
| Gr.Dino-SAM | 10.48 | 32.33 | 46.04 |
| LISA | 14.90 | 56.03 | 75.79 |
| DeiSAM (ours) | **65.14** | **85.40** | **87.83** |

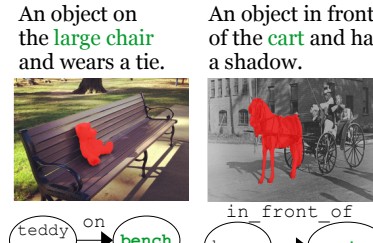

Figure 4: **DeiSAM handles ambiguous prompts.** Results with prompts (top) with scene graphs (bottom).

We compare DeiSAM to multiple purely neural approaches, both empirically as well as qualitatively. We include three baselines that use one model for object identification and subsequently segment the grounded image using SAM (Kirillov et al., 2023), similar to the grounding in DeiSAM. Our comparison includes the following models for visual grounding: 1) One-For-All (OFA) (Wang et al., 2022), a unified transformer-based sequence-to-sequence model for vision and language tasks of which we use a dedicated visual grounding checkpoint[5]. 2) Grounded Language-Image Pre-training (GLIP) (Li et al., 2022b) a model for specifically designed for object-aware and semantically-rich object detection and grounding. 3) GroundingDino (Liu et al., 2023c) an open-set object detector combining transformer-based detection with grounded pre-training. Moreover, we compare to an end-to-end semantic segmentation model supporting textual prompts with SEEM (Zou et al., 2023). Lastly, we compare to LISA (Lai et al., 2023), a state-of-the-art neural reasoning segmentation model.

## 5.2 Empirical Evidence

The results on DeiVG of all baselines compared to DeiSAM are summarized in Tab. 1. DeiSAM clearly outperforms all purely neural approaches by a large margin on all splits of DeiVG. The performance of most methods improves with more descriptive deictic prompts, *i.e.*, more relations being used. We attribute this effect to two distinct causes. For one, additional information describing the target object contributes to higher accuracy in object detection. On the other hand, DeiVG$_1$ contains significantly more samples with multiple target objects than DeiVG$_2$ or DeiVG$_3$. Consequently, cases in which a method identifies only one out of multiple objects will have a higher impact on the overall performance. Overall, the large gap between DeiSAM and all baselines highlights the lack of complex reasoning capabilities in prevalent models and DeiSAM's large advantage. We further provide a runtime comparison and its analysis in App. F, showcasing that DeiSAM's runtime is comparable to the baselines, and the bottleneck is in the LLMs, not in the reasoning pipeline.

## 5.3 Qualitative Evaluation

After empirically demonstrating DeiSAM's capabilities, we look into some qualitative examples. In Fig. 4, we demonstrate the efficacy of the semantic unifier. All examples use terminology in the deictic prompt diverging from the scene graph entity names. Nonetheless, the unification step successfully maps synonymous terms and still produces the correct segmentation masks, overcoming the limitation of off-the-shelf symbolic logic reasoners.

In Fig. 5, we further compare DeiSAM with the purely neural baselines. DeiSAM produces the correct segmentation mask even for complicated shapes (*e.g.*, partially occluded cable) or complex scenarios (*e.g.*, multiple people, only some holding umbrellas). All baseline methods, however, regularly fail to identify the correct object. A common failure mode is confounding nouns in the deictic prompt. For example, when describing an object in relation to a 'boat', the boat itself is identified instead of the target object. These examples strongly illustrate the improvements of DeiSAM over the pure neural approach on abstract reasoning tasks.

---

[5] https://modelscope.cn/models/damo/ofa_visual-grounding_refcoco_large_en/summary

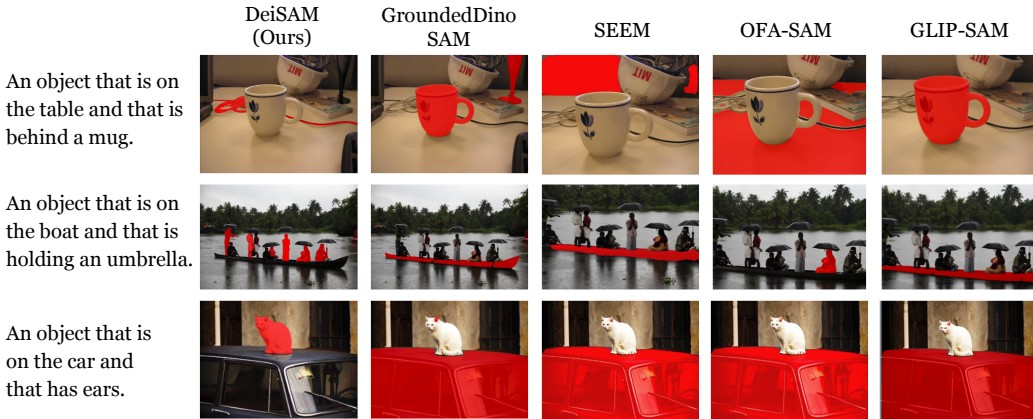

Figure 5: **DeiSAM segments objects with deictic prompts.** Segmentation results on the DeiVG dataset using DeiSAM and baselines are shown with deictic prompts. DeiSAM correctly identifies and segments objects given deictic prompts (left-most column), while the baselines often segment a wrong object. More results are available in App. G (Best viewed in color).

## 5.4 Ablations

The modular nature of the DeiSAM pipeline enables easy component variations. Next, we investigate the performance of key modules in isolation and their overall influence on the pipeline.

**LLM Rule Generation.** One of the key steps for DeiSAM is the translation of deictic prompts posed in natural language into syntactically and semantically sound logic rules. We observed that the performance of instruction-tuned LLMs on this task heavily depends on the employed prompting technique. Consequently, we leverage the well-known methods of few-shot prompting (Brown et al., 2020) and chain-of-thought (CoT) (Wei et al., 2022).

To that end, we first let the model extract all predicates from a deictic prompt, which we sub-

Table 2: Ablations on prompting techniques for rule generation w/ Llama-2-13B-Chat. Few-shot examples are imperative for rule generation with chain-of-thought (CoT) prompting providing additional improvements for complex deictic prompts.

| Prompting | Overall Success (%) ↑ | | |
|---|---|---|---|
| Technique | $DeiVG_1$ | $DeiVG_2$ | $DeiVG_3$ |
| Instruct Only | 0.00 | 0.00 | 0.00 |
| CoT | 0.00 | 0.00 | 0.00 |
| Few-shot | **94.04** | 92.52 | 90.17 |
| Few-shot + CoT | 91.00 | **95.17** | **93.45** |

sequently provide as additional context for the rule generation. For both cases, we provide multiple few-shot examples. We evaluate all prompting approaches with LLama-2-13B (Touvron et al., 2023) in Tab. 2. Clearly, few-shot examples are imperative to perform rule generation successfully. Additionally, CoT for predicate decomposition further improves the rule generation for complex prompts.

With the best prompting technique identified, we additionally evaluated multiple open and closed-source language models of different sizes (*cf.* App. F). In general, all instruction-tuned models can generate logic rules from deictic prompts. However, larger models strongly outperform smaller ones, especially for more complex inputs. The overall best-performing model was gpt-3.5-turbo producing correct rules for DeiVG for 93.65% of all samples.

**Semantic Unification.** Next, we take a more detailed look into the semantic unification module. At this step, we bridge the semantic gap between differing formulations in the deictic prompt and the scene graph generator. To evaluate this task, we created an exemplary benchmark based on synonyms in the visual genome. For 2.5k scenes in DeiVG, we considered all objects in the scene graph and identified one object name that differed from its synset entry. Based on that synonym, the task is to identify the one, unique, synonymous object in the scene. For example, in an image containing a 'table', 'couch', 'chair', and 'cupboard' the query 'sofa' should identify the 'couch' as most likely synoynm. Overall, the task is considerably more challenging than it may appear at first glance, with the best model only achieving a success rate of 72% (*cf.* App F). We observed, for example, the query 'sofa' is matched with 'pillow' instead of the targeted 'couch' or 'trousers' with 'jacket' instead of 'pants'. These results motivate further research into the semantic unification process.

Table 3: Comparison on RefCOCO+.

| Method | Mean Average Precision (%) ↑ | | |
| --- | --- | --- | --- |
| | val | testA | testB |
| LISA | 67.55 | 74.86 | 63.03 |
| GroundedSAM | 55.09 | 66.21 | 44.21 |
| DeiSAM | **71.72** | **77.29** | **64.98** |

Table 4: Comparison on DeiRefCOCO+.

| Method | Mean Average Precision (%) ↑ | | |
| --- | --- | --- | --- |
| | val | testA | testB |
| LISA | 44.92 | 47.60 | 43.23 |
| GroundedSAM | 30.06 | 31.75 | 28.12 |
| DeiSAM | **71.56** | **79.51** | **66.43** |

## 5.5 Solving Reference Expression

In addition to experiments on DeiVG, we also consider the RefCOCO dataset (Yu et al., 2016), which comprises referring expressions for object segmentation. Thus, this dataset is the most similar setup to the deictic segmentation task in prior work. The key difference between RefCOCO and DeiVG is that the latter is built to evaluate models' *abstract* reasoning capabilities in complex visual scenes. In contrast, RefCOCO mainly evaluates descriptive object identifications, *i.e.*, objects with names and properties, *e.g.* "old man or child in green short", compared to *e.g.* "an object on a table and next to a computer" in DeiVG. Consequently, deictic prompts are more challenging than the reference texts in RefCOCO, since DeiVG prompts do *not* include explicit names and properties of target objects.

Since there were no publicly available scene graphs (or SGGs) for MSCOCO images, we used GPT3 to convert the reference text to a structured scene graph representation with additional annotations. Tab. 3 shows the mAP of LISA, GroudnedSAM, and DeiSAM on RefCOCO. LISA achieved better overall performances than GroundedSAM, showing its strong capability on the reference task. DeiSAM, however, remains competitive with LISA and achieves better results on all splits.

To further investigate the challenging nature of abstract prompts, we curated a DeiRefCOCO+ benchmark that contains more abstract textual references. Specifically, we turned the reference texts in RefCOCO+ into deictic prompts by removing any description of the target object. For example, the prompt "kid wearing navy shirt" is modified to "an object that is wearing navy shirt". Tab. 4 again shows the mAP of DeiSAM, GroundedSAM, and LISA on the modified DeiRefCOCO+ dataset. Importantly, DeiSAM retains a similar performance on both types of prompt formulations. In comparison, we observe a strong drop in performance for GroundedSAM and LISA with the absence of confounding object descriptions. These results further highlight the strength and abstraction level of the performed reasoning performed by DeiSAM [6].

## 5.6 DeiCLEVR – Abstract Reasoning Segmentation

DeiSAM excels in high-level abstract reasoning, where purely neural pipelines often struggle. To demonstrate, we developed DeiCLEVR, an abstract reasoning segmentation task based on CLEVR (Johnson et al., 2017). This task challenges models with abstract concepts and relationships.

**Task.** The task is to segment objects given prompts where the answers are derived by the reasoning over abstract list operations. We consider 2 operations: delete and sort. The input is a pair of an image and a prompt, e.g. "Segment the second left-most object after deleting a gray object?". Examples are shown in Fig. 6. To solve this task, models need to understand the visual scenes and perform high-level abstract reasoning to segment.

**Dataset. (Image)** Each scene contains at most 3 objects with different attributes: (i) colors of cyan, gray, red, and yellow, (ii) shapes of sphere, cube, and cylinder, (iii) materials of metal and matte. We excluded color duplications in a single image. **(Prompts)** We generated prompts using a templates: "The [Position] object after [Operation]?", where [Position] can take either of: left-most first, second, or third. [Operation] can take either of: (I) delete an object, and (II) sort the objects in the order of: cyan < gray < red < yellow (alphabetical order). We generated 10k examples for each operation.

**Models.** We used DeiSAM with Slot Attention (Locatello et al., 2020) pretrained on the visual Inductive Logic Programming (ILP) dataset (Shindo et al., 2024), which contains positive and negative visual scenes for list operations. We used GroundedSAM and LISA for neural baselines (*cf.* App. E).

---

[6]In App. F, we demonstrate experiments on DeiVG with prompts in the RefCOCO's reference format, highlighting the robustness of DeiSAM against neural baselines.

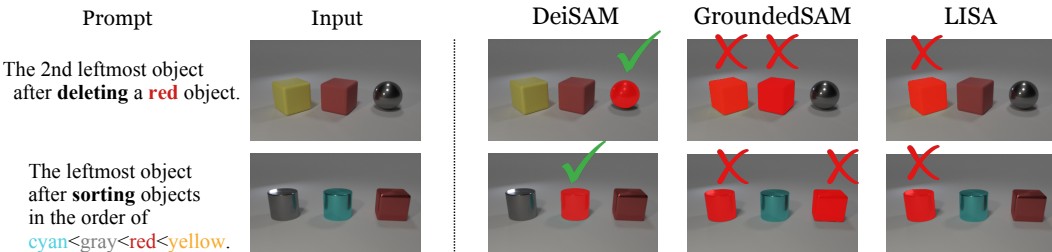

| Prompt | Input | DeiSAM | GroundedSAM | LISA |
|--------|-------|--------|-------------|------|

Figure 6: **DeiSAM performs abstract reasoning segmentation.** When presented with a visual scene paired with an abstract, complex prompt (left), DeiSAM effectively identifies and segments the object specified by the prompt, while neural baselines frequently fail to deduce the target object (right).

**Result.** In Table 5, we present the mean Average Precision (mAP) for each baseline evaluated. The purely neural baselines struggle to accurately deduce segmentations in response to abstract reasoning prompts, while DeiSAM excels at identifying and segmenting the object specified by the prompt.

Moreover, Fig. 6 provides qualitative examples illustrating that DeiSAM effectively segments objects requiring high-level reasoning. In contrast, the neural baselines frequently fail to segment the correct target object. These findings indicate that existing neural baselines are inadequate for addressing abstract reasoning prompts. We demonstrate that integrating differentiable logic reasoners can significantly enhance reasoning capabilities.

Table 5: **DeiSAM handles abstract visual reasoning.** mAP on DeiCLEVR.

| mAP ( ↑ ) | Delete | Sort |
|-----------|--------|------|
| DeiSAM | **99.29** | **99.57** |
| GroundedSAM | 7.6 | 15.39 |
| LISA | 12.88 | 11.15 |

## 5.7  End-to-End Training of DeiSAM

Since DeiSAM employs a differentiable forward reasoner, a meaningful gradient signal can be back-propagated through the entire pipeline. Consequently, DeiSAM enables end-to-end learning on complex object detection and segmentation tasks with logical reasoning explicitly modeled during training. To illustrate this property, we show that DeiSAM can learn weighted mixtures of scene graph generators by propagating gradients through the reasoning module.

**Task.** We consider 2 distinct scene graph generators and compose a weighted mixture of them. We show an example for the deictic prompt "An object that has hair and that is on a surfboard" in Listing 2. The first 3 rules compute the target object for each SGG, similarly to `Program 1`, and the last 2 rules produce a weighted merge

```
// Program 2
targetSgg(X,SG):-cond1(X,SG),cond2(X,SG).
cond1(X,SG):-hasSgg(X,Y,SG),typeSgg(Y,hair,SG).
cond2(X,SG):-onSgg(X,Y,SG),onSgg(Y,surfboard,SG).
// Compose weighted mixtures.
w_1: target(X):-targetSgg(X,sgg1).
w_2: target(X):-targetSgg(X,sgg2).
```

Listing 2: A program for SGG learning.

of both predictions. Importantly, `Program 2` utilizes different SGGs (*i.e.* variable `SG`) and merges the results using learnable weights (*i.e.* `w_1` and `w_2`). Consequently, the learning task is the optimization of weights $w_i \in \mathbb{R}$ for downstream deictic segmentation. The differentiability of the DeiSAM pipeline allows efficient gradient-based optimizations.

**Experimental Setup.** We used VETO (Sudhakaran et al., 2023), which outperforms other SGGs on biased datasets where only some relations appear frequently. As the second 'SGG' for our weighted mixture model, we relied on ground-truth scene graphs from Visual Genome. We consider the following baselines: DeiSAM-VETO that only uses a pre-trained VETO model (Sudhakaran et al., 2023), DeiSAM-Mixture (naive) that uses a mixture of VETO and VG scene graphs with randomly initialized weights. We compare those approaches to DeiSAM-Mixture*, which uses the trained mixture. We extracted instances from DeiVG datasets not used in VETO training (ca. 2000 samples), which we divided into a training, validation, and test split. For rule generation, we use the same system prompt and models as in Sec. 5.1, adapting the generated programs for weight learning.

We minimize the binary cross entropy loss with respect to rule weights `w_1` and `w_2`. To calculate this loss, we provide labels for predicted masks in the model, *i.e.*, a binary label $y_i \in \{0, 1\}$. For each instance in DeiVG, DeiSAM predicts segmentation masks in the forward pass, and gradients are backpropagated through the differentiable forward reasoner (*cf.* App. E.1 for details).

Table 6: **End-to-end training improves DeiSAM.** Mean Average Precision on the test split of the task of learning SGGs. DeiSAM-VETO uses a trained VETO model (Sudhakaran et al., 2023), DeiSAM-Mixture (naive) uses a mixture of a trained VETO model and VG scene graphs with randomly initialized rule weights, DeiSAM-Mixture* uses the resulted mixture model after the weight learning.

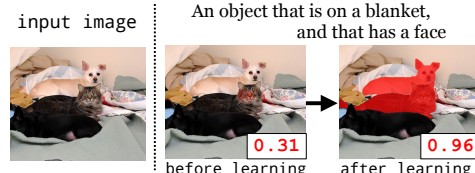

Figure 7: **DeiSAM can learn to produce better masks.** Shown are the input image (left) and target segmentation masks together with confidence scores obtained before (middle) and after (right) end-to-end training DeiSAM. DeiSAM improves the quality of segmentation by learning (Best viewed in color).

|  | mAP (%) ↑ | |
| --- | --- | --- |
| Method | DeiVG$_1$ | DeiVG$_2$ |
| DeiSAM-VETO | 6.64 | 15.92 |
| DeiSAM-Mixture (naive) | 37.61 | 59.81 |
| DeiSAM-Mixture* | **64.44** | **86.57** |

**Result.** In Tab. 6, we compare the mAP on the test split. The trained model DeiSAM-Mixture* clearly outperforms the naive baseline, demonstrating successful training of the DeiSAM pipeline using gradients via differentiable reasoning. DeiSAM-VETO weak performance can be attributed to objects that appear only on prompts but not its training data (*cf.* App. D).

Fig 7 shows examples of segmentation masks and their confidence scores produced by DeiSAM-Mixture models before and after training. Before learning, wrong or incomplete regions are segmented with low confidence scores because the reasoner fails to identify correct objects with low-quality scene graphs that miss critical objects and relations. After learning, DeiSAM produces faithful segmentation masks and increased confidence scores. This experiment highlights that DeiSAM improves the quality of scene graphs and the subsequent segmentation masks by learning using gradients, *i.e.*, it is a fully trainable pipeline with a strong capacity for complex logic reasoning.

## 6 Conclusion

Before concluding, let us discuss the limitations and future research directions. Our investigation of DeiSAM's components highlights some clear avenues for future research. While LLMs perform well at parsing deictic prompts into logic rules with few-shot prompting, their performance could be improved further by, *e.g.*, syntactically constrained sampling[7] or dedicated fine-tuning. Further, the observed challenges in semantic unification could be addressed by querying LLMs instead of using embedding models or providing multiple weighted candidates to the reasoner.

Upon manual inspection of the DeiVG dataset, we identified some inconsistent examples annotated with erroneous scene graphs in Visual Genome that cannot be automatically cleaned up without external object identification (*cf.* App. D). Our results support the assessment that generating rich scene graphs is key but difficult to achieve in a zero-shot fashion. However, as we demonstrated, the differentiable pipeline of DeiSAM can be utilized for meaningful training on complex downstream tasks. Thus allowing for the incorporation of real-world use cases in the training of SGGs in an end-to-end fashion. Further, DeiSAM can provide valuable information on general performance and failure cases of SGGs by investigating deictic segmentation tasks. Furthermore, the modularity of DeiSAM allows for easy integration of potential improvements to any of its components.

To conclude, we proposed DeiSAM to perform deictic object segmentation in complex scenes. DeiSAM effectively combines large-scale neural networks with differentiable forward reasoning in a modular pipeline. DeiSAM allows users to intuitively describe objects in complex scenes by their relations to other objects. Moreover, we introduced the novel Deictic Visual Genome (DeiVG) benchmark for segmentation with complex deictic prompts. In our extensive experiments, we demonstrated that DeiSAM strongly outperforms neural baselines highlighting its strong reasoning capabilities on visual scenes with complex textual prompts. To this end, our empirical results revealed open research questions and important future avenues of visual scene understanding.

---

[7] https://github.com/IsaacRe/Syntactically-Constrained-Sampling

**Acknowledgements.** The authors thank Maurice Kraus and Felix Friedrich for their valuable feedback on the manuscript. This work was partly supported by the EU ICT-48 Network of AI Research Excellence Center "TAILOR" (EU Horizon 2020, GA No 952215), and the Collaboration Lab "AI in Construction" (AICO). The work has also benefited from the Federal Ministry of Education and Research (BMBF) Competence Center for AI and Labour ("KompAKI", FKZ 02L19C150) and from the Hessian Ministry of Higher Education, Research, Science and the Arts (HMWK) cluster projects "The Third Wave of AI" and "The Adaptive Mind". We gratefully acknowledge support by the German Center for Artificial Intelligence (DFKI) project "SAINT". This work also benefited the HMWK / BMBF ATHENE project "AVSV" and the National High Performance Computing Center for Computational Engineering Science (NHR4CES). The Eindhoven University of Technology authors received support from their Department of Mathematics and Computer Science and the Eindhoven Artificial Intelligence Systems Institute.

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

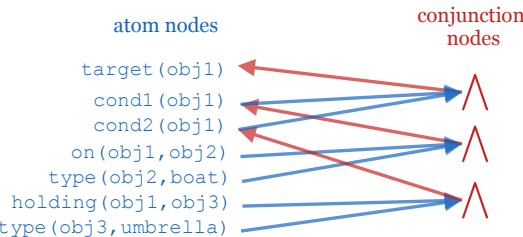

Figure 8: Forward reasoning graph for `Program 1` in Listing 1. A reasoning graph consists of *atom nodes* and *conjunction nodes*, and is obtained by grounding rules *i.e.*, removing variables by, *e.g.*, X ← obj1, Y ← obj2. By performing bi-directional message passing on the reasoning graph using soft-logic operations, DeiSAM computes logical consequences in a differentiable manner. Only relevant nodes are shown (Best viewed in color).

## A    First-Order Logic and Differentiable Reasoning.

We provide a formal definition of first-order logic (FOL). An *atom* is a formula $p(t_1, \ldots, t_n)$, where $p$ is a predicate symbol (*e.g.* type) and $t_1, \ldots, t_n$ are terms. A term is a variable or a constant. A *ground atom* or simply a *fact* is an atom with no variables (*e.g.* type(obj1, boat)). A *literal* is an atom ($A$) or its negation ($\neg A$), and a *clause* is a finite disjunction ($\vee$) of literals. A *definite clause* is a clause with exactly one positive literal. If $A, B_1, \ldots, B_n$ are atoms, then $A \vee \neg B_1 \vee \ldots \vee \neg B_n$ is a definite clause. We write definite clauses in the form of $A$ :- $B_1, \ldots, B_n$, and refer to them as *rules* for simplicity in this paper. *Forward Reasoning* is a data-driven approach of reasoning in FOL (Russell & Norvig, 2010), *i.e.*, given a set of facts and a set of rules, new facts are deduced by applying the rules to the facts. Differentiable forward reasoners compute logical entailment using tensor representations (Evans & Grefenstette, 2018; Shindo et al., 2023) or graph neural networks (Shindo et al., 2024), and perform rule learning using gradients given labeled examples in the form of inductive logic programming (Cropper & Dumancic, 2022).

DeiSAM employs a graph neural network-based differentiable forward reasoner (Shindo et al., 2024), and we briefly explain the reasoning process. We represent a set of (weighted) rules as a directed bipartite graph. For example, Fig. 8 is a reasoning graph that represents `Program 1`.

### A.1    Details of Differentiable Forward Reasoning

We provide the details of differentiable forward reasoning.

**Definition A.1.** A *Forward Reasoning Graph* is a bipartite directed graph $(\mathcal{V}_\mathcal{G}, \mathcal{V}_\wedge, \mathcal{E}_{\mathcal{G} \rightarrow \wedge}, \mathcal{E}_{\wedge \rightarrow \mathcal{G}})$, where $\mathcal{V}_\mathcal{G}$ is a set of nodes representing ground atoms (atom nodes), $\mathcal{V}_\wedge$ is set of nodes representing conjunctions (conjunction nodes), $\mathcal{E}_{\mathcal{G} \rightarrow \wedge}$ is set of edges from atom to conjunction nodes and $\mathcal{E}_{\wedge \rightarrow \mathcal{G}}$ is a set of edges from conjunction to atom nodes.

DeiSAM performs forward-chaining reasoning by passing messages on the reasoning graph. Essentially, forward reasoning consists of *two* steps: (1) computing conjunctions of body atoms for each rule and (2) computing disjunctions for head atoms deduced by different rules. These two steps can be efficiently computed on bi-directional message-passing on the forward reasoning graph. We now describe each step in detail.

**(Direction →) From Atom to Conjunction.** First, messages are passed to the conjunction nodes from atom nodes. For conjunction node $v_i \in \mathcal{V}_\wedge$, the node features are updated:

$$v_i^{(t+1)} = \bigvee \left( v_i^{(t)}, \bigwedge_{j \in \mathcal{N}(i)} v_j^{(t)} \right),  \tag{1}$$

where $\bigwedge$ is a soft implementation of *conjunction*, and $\bigvee$ is a soft implementation of *disjunction*. Intuitively, probabilistic truth values for bodies of all ground rules are computed softly by Eq. 1.

**(Direction ←) From Conjunction to Atom.** Following the first message passing, the atom nodes are then updated using the messages from conjunction nodes. For atom node $v_i \in \mathcal{V}_\mathcal{G}$, the node features are updated:

$$v_i^{(t+1)} = \bigvee \left( v_i^{(t)}, \bigvee_{j \in \mathcal{N}(i)} w_{ji} \cdot v_j^{(t)} \right),  \tag{2}$$

where $w_{ji}$ is a weight of edge $e_{j\to i}$. We assume that each rule $C_k \in \mathcal{C}$ has its weight $\theta_k$, and $w_{ji} = \theta_k$ if edge $e_{j\to i}$ on the reasoning graph is produced by rule $C_k$. Intuitively, in Eq. 2, new atoms are deduced by gathering values from different ground rules and from the previous step.

We used product for conjunction, and *log-sum-exp* function for disjunction:

$$softor^{\gamma}(x_1, \ldots, x_n) = \gamma \log \sum_{1 \leq i \leq n} \exp(x_i/\gamma), \tag{3}$$

where $\gamma > 0$ is a smooth parameter. Eq. 3 approximates the maximum value given input $x_1, \ldots, x_n$.

**Prediction.** The probabilistic logical entailment is computed by the bi-directional message-passing. Let $\mathbf{x}_{atoms}^{(0)} \in [0, 1]^{|\mathcal{G}|}$ be input node features, which map a fact to a scalar value, RG be the reasoning graph, $\mathbf{w}$ be the rule weights, $\mathcal{B}$ be background knowledge, and $T \in \mathbb{N}$ be the infer step. For fact $G_i \in \mathcal{G}$, DeiSAM computes the probability as:

$$p(G_i \mid \mathbf{x}_{atoms}^{(0)}, \mathsf{RG}, \mathbf{w}, \mathcal{B}, T) = \mathbf{x}_{atoms}^{(T)}[i], \tag{4}$$

where $\mathbf{x}_{atoms}^{(T)} \in [0, 1]^{|\mathcal{G}|}$ is the node features of atom nodes after $T$-steps of the bi-directional message-passing.

By optimizing the cross-entropy loss, the differentiable forward reasoner can solve Inductive Logic Programming (ILP) problems with propositional encoding (Shindo et al., 2018), where the task is to find classification rules given positive and negative examples. It has been extensively applied to solve complex visual patterns (Helff et al., 2023), image generation (Deiseroth et al., 2022), meta-level reasoning and learning (Ye et al., 2022), predicate invention (Sha et al., 2024, 2023), and self-explanatory learning (Stammer et al., 2024).

## B System Prompts for Rule Generation

To generate logic rules using LLMs, we used the following system prompt.

```
Given a deictic representation and available predicates, generate rules in the format.
target(X):-cond1(X),...condn(X).
cond1(X):-pred1(X,Y),type(Y,const1).
...
condn(X):-predn(X,Y),type(Y,const2).
Use predicates and constants that appear in the given sentence.
Capitalize variables: X, Y, Z, W, etc.
```

In practice, this system prompt combined with few-shot examples for a downstream task (*cf.* App. E).

## C DeiVG Datasets

We generated DeiVG dataset using Visual Genome dataset (Krishna et al., 2017). We used the entire Visual Genome dataset to generate deictic prompts and answers out of scene graphs, and we randomly downsampled 10k examples. We only considered the following relations:

- 'on'
- 'wears'
- 'has'
- 'parked on'
- 'behind'
- 'holding'
- 'against'
- 'wearing'
- 'near'
- 'along'
- 'in front of'
- 'at'
- 'under'
- 'sitting on'
- 'made of'
- 'above'
- 'carrying'
- 'riding'
- 'over'

The prompts are synthetically generated by extracting relations that shares the same subject in the scene. For example, with a pair of VG relations, *"person is holding an umbrella"* and *"person on a boat"*, we generate a deictic prompt, *"an object that is holding an umbrella, and that is on a boat"*. Subsequently, the corresponding answer is extracted from the scene graph. We provide two instances in the generated DeiVG$_2$ dataset in Listing 3.

```
% Example 1
{
    deictic_prompt: "an object that is on a sofa, and that is on a book",
    answer: [{"name": "paper",
            "h": 32,
            "synsets": ["paper.n.01"],
            "object_id": 2687751,
            "w": 61,
            "y": 236,
            "x": 272}],
    VG_image_id: 2376540,
    VG_data_index: 44297]
}
% Example 2
{
    deictic_prompt: "an object that has a shadow, and that is wearing a black shirt",
    answer: [{"h": 50,
            "object_id": 1001275,
            "merged_object_ids": [1001270],
            "synsets": ["man.n.01"],
            "w": 21, "y": 333,
            "x": 47,
            "names": ["man"]}],
    VG_image_id: 2365153,
    VG_data_index: 55196]
}
```

Listing 3: DeiVG examples.

## D   Additional Analysis on VG Scene Graphs

We provide a further investigation of Visual Genome (VG) scene graph annotations. We demonstrate that (1) VG annotations have multiple versions and there is a non-trivial discrepancy between their relation distributions, and (2) VG annotations contain incomplete and erroneous scene graphs that cannot be automatically cleaned up without external object identification.

**VG Annotation Discrepancy**   We investigated the scene graph generator module, going beyond the ground truth scene graphs of Visual Genome. One of the key challenges in using a pre-trained SGG is the potential mismatch between the set of objects and annotations in DeiVG and the training data of the SGG. While we built on the latest version of Visual Genome with extended annotations ($VG_{v1.4}$), an older version (Krishna et al., 2017) has been commonly used for benchmarking different SGGs by excluding non-frequent object types and relations (Xu et al., 2017). This preprocessed older version (VG) is still considered the standard benchmark for SGGs, which lacks many crucial objects and attributes in DeiVG built upon $VG_{v1.4}$. We illustrate the discrepancy in Fig. 9, where we compare the Kernel Density Estimate (KDE) of $VG_{v1.4}$ and VG. It shows that the newer version contains more types of objects in the top-and-middle frequency range. Additionally, many object types of $VG_{v1.4}$ in the middle-frequency range are not contained at all in VG. Consequently, when parsing scenes from DeiVG with SGGs pre-trained on VG, the objects in the deictic prompt may not be contained in the generated scene graph at all. For example, for a given deictic prompt "an object that is wearing a black shirt", we observed that the pre-trained SGG failed to detect *black shirt* because it was not included in its training data. This discrepancy leads to sub-par performance of DeiSAM with a pre-trained SGG, as shown in Tab. 6. While training the SGG specifically on the relevant scene distribution can partially address this issue, it is crucial to highlight that DeiSAM can be leveraged to improve SGGs as well. Our subsequent demonstration in Section 5.7 provides a glimpse of DeiSAM's capabilities, showcasing its differentiable forward reasoner's ability to perform end-to-end training, thereby unlocking new horizons in scene understanding and reasoning.

In Fig. 10, the plot on the left depicts the distribution of the least frequent tail object types on the latest extended version ($VG_{v1.4}$) in comparison to the older version (VG) (Krishna et al., 2017). Object types are sorted with respect to the number of occurrences (x-axis). In the non-frequent range of 0 to 1000, $VG_{v1.4}$ contains object types that never appear on the older version (VG), revealing these tail classes of object types are completely missing in the processed older annotations (Xu et al., 2017),

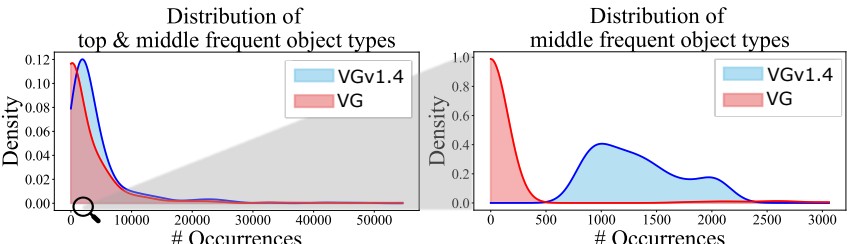

Figure 9: **The large discrepancy between DeiVG and standard Visual Genome.** DeiVG uses Visual Genome with extended annotations ($VG_{v1.4}$), but the older version (VG) (Krishna et al., 2017) has been commonly used for SGG training. Object types are sorted w.r.t. the number of occurrences (x-axis) and their corresponding occupancies are shown (y-axis). The left plot is for top-and-middle frequent object types (0-50k #occ.), and the right plot is for middle frequent object types (0-3k #occ.). $VG_{v1.4}$ contains many object types that do not appear in VG (Best viewed in color).

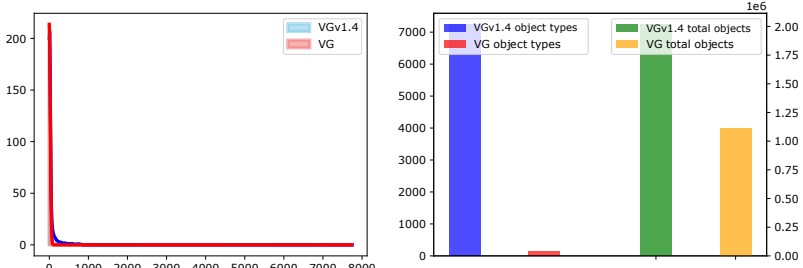

Figure 10: Further comparisons of $VG_{v1.4}$ with VG. The plot on the left depicts the distribution of the least frequent tail object types of the latest extended version ($VG_{v1.4}$) in comparison to the older version (VG) (Krishna et al., 2017). Object types are sorted w.r.t. the number of occurrences (x-axis). The plot on the right shows the comparison of the number of object types and the number of total objects in the datasets. $VG_{v1.4}$ contains many more object types than VG, making it difficult for SGGs that are pre-trained on the older version to achieve high performance (Best viewed in color).

which are commonly used for benchmarking SGGs. Moreover, the bar plot on the right indicates a notable increase in the number of object types and total object counts in $VG_{v1.4}$ in comparison to VG, making it difficult for SGGs that are pre-trained on the older version to achieve high performance on the DeiVG dataset built upon $VG_{v1.4}$.

**Errors on VG Annotations.** Upon manual inspection of the DeiVG dataset, we identified some inconsistent examples resulting from incomplete and erroneous scene graphs in Visual Genome that cannot be automatically cleaned up without external object identification. For example, the annotations shown in Fig. 11 lead to a DeiVG sample with two missing target objects and an incorrect one. We plan on building a more consistent deictic segmentation benchmark using a cleanup process similar to GQA (Hudson & Manning, 2019).

# E   Details of Experiments

We provide details of the models used in the evaluation. For all methods using SAM for segmentation—including DeiSAM—we use the same publicly available SAM checkpoint[8].

**DeiSAM.** We used NEUMANN (Shindo et al., 2024) with $\gamma = 0.01$ for soft-logic operations, and the number of inference steps is set to 2. We set the box threshold to 0.3 and the text threshold to 0.25 for the SAM model. All generated rules are assigned a weight of 1.0. If no targets are detected, DeiSAM produces a mask of a randomly chosen object in the scene.

For LLMs, we provided few-shot examples of deictic prompts and desired outputs as shown in Listing 4. These few-shot examples improved the quality of the rule generation that follows a certain format. These are combined with the system prompt in App. B to generate rules by LLMs.

---

[8] https://huggingface.co/spaces/abhishek/StableSAM/blob/main/sam_vit_h_4b8939.pth

Prompt: *'An object that is wearing a helmet'*

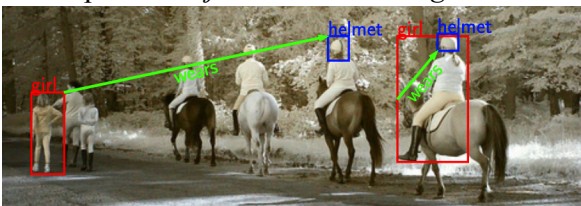

Figure 11: Example of erroneous annotations in Visual Genome leading to inconsistent examples in DeiVG. Here, the annotations for 'helmet' are incomplete and in one case, the relation 'wears' is linked to the wrong person (Best viewed in color).

```
Examples:

an object that is next to a keyboard.
available predicates: next_to
cond1(X):-next_to(X,Y),type(Y,keyboard).
target(X):-cond1(X).

an object that is on a desk.
available predicates: on
cond1(X):-on(X,Y),type(Y,desk).
target(X):-cond1(X).

an object that is on a ground, and that is behind a white line.
available predicates: on,behind
cond1(X):-on(X,Y),type(Y,ground).
cond2(X):-behind(X,Y),type(Y,whiteline).
target(X):-cond1(X),cond2(X)

an object that is near a desk and against wall.
available predicates: near,against
cond1(X):-near(X,Y),type(Y,desk).
cond2(X):-against(X,Y),type(Y,wall).
target(X):-cond1(X),cond2(X).

an object that has sides, that is on a pole, and that is above a stop sign.
available predicates: has,on,above
cond1(X):-has(X,Y),type(Y,sides).
cond2(X):-on(X,Y),type(Y,pole).
cond3(X):-above(X,Y),type(Y,stopsign).
target(X):-cond1(X),cond2(X),cond3(X).

an object that is wearing a shirt, that has a hair, and that is wearing shoes.
available predicates: wearing,has,wearing
cond1(X):-wearing(X,Y),type(Y,shirt).
cond2(X):-has(X,Y),type(Y,hair).
cond3(X):-wearing(X,Y),type(Y,shoes).
target(X):-cond1(X),cond2(X),cond3(X).
```

Listing 4: Few-short examples for rule generation using LLMs.

**GroundedSAM.** We used a publicly available GroundedDino version[9] with Swin-B backbone, pre-trained on COCO, O365, GoldG, Cap4M, OpenImage, ODinW-35 and RefCOCO. We set the box threshold to $0.3$ and the text threshold to $0.25$ for the SAM model.

**GLIP** We used a publicly available version of GLIP-L[10] with a Swin-L backbone and pre-trained on FourODs, GoldG, CC3M+12M, SBU. We set the prediction threshold to $0.4$.

**OFA** We used a publicly available version of OFA-Large[11] fine-tuned for visual grounding.

---

[9] https://github.com/IDEA-Research/GroundingDINO/releases/download/v0.1.0-alpha2/groundingdino_swinb_cogcoor.pth

[10] https://huggingface.co/GLIPModel/GLIP/blob/main/glip_large_model.pth

[11] https://modelscope.cn/models/iic/ofa_visual-grounding_refcoco_large_en/summary

**SEEM** We used a publicly available checkpoint of SEEM[12] with Focal-L backbone.

**Evaluation Metric.** We used mean average precision (mAP) to evaluate segmentation models. Segmentation masks are converted to corresponding bounding boxes by computing their contours, and then mAP is computed by comparing them with the ground truth bounding boxes provided by Visual Genome.

### E.1 Learning to Segment Better

We describe the experimental setting in more detail.

**Method.** Let $(\boldsymbol{x}, p) \in \mathcal{D}$ be a DeiVG dataset, where $\boldsymbol{x}$ is an input image, and $p$ is a deictic prompt. For each input, DeiSAM produces segmentation masks $\mathbf{M}_i$ with corresponding confidence score $s_i$, *i.e.*, $\{(\mathbf{M}_i, s_i)\}_{i=0,...n} = f_{deisam}(\boldsymbol{x}, p)$. For each mask $\mathbf{M}_i$, we consider a binary label $y_i \in \{0, 1\}$ using its corresponding bounding box and comparing with ground-truth masks with the IoU score, which assesses the quality of bounding boxes. For each instance $(\boldsymbol{x}, p) \sim \mathcal{D}$, we compute the binary-cross entropy loss: $\ell = -\sum_{(\mathbf{M}_i, s_i) \sim f_{deisam}(\boldsymbol{x}, p)} (y_i \log s_i + (1 - y_i) \log(1 - s_i))$, We minimize the loss $\ell$ through gradient descent with respect to the rule weights in the differentiable programs.

**Task.** Given SGGs $sgg_1, sgg_2, \ldots, sgg_n$, we compose a DeiSAM model with a mixture of them, *i.e.* $sgg(\boldsymbol{x}) = \sum_i w_i sgg_i(\boldsymbol{x})$ with $w_i \in [0, 1]$ and $sgg_i : \mathbb{R}^2 \to [0, 1]^{|\mathcal{G}|}$, where $\mathcal{G}$ is a set of facts to describe scene graphs. For example, `Program 2` shown on the right represents the mixture of scene graph generators for the deictic prompt "An object that has hair and that is on a surfboard". In contrast to `Program 1`, the program utilizes different SGGs and merges the results using learnable weights. The learning task is the optimization of weights $w_i$ for downstream deictic segmentation. The differentiable reasoning pipeline in DeiSAM allows efficient gradient-based optimizations with automatic differentiation.

**Dataset.** Let $(\boldsymbol{x}, p) \in \mathcal{D}$ be a DeiVG dataset, where $\boldsymbol{x}$ is an input image, and $p$ is a deictic prompt and $(o_1, \ldots, o_m) \in \mathcal{A}$ be the answers, where $o_1, \ldots o_m$ are correct target objects to be segmented specified by the prompt. For each input, DeiSAM produces segmentation masks with their confidence:

$$\{(\mathbf{M}_i, s_i)\}_{i=0,...n} = f_{deisam}(\boldsymbol{x}, p), \tag{5}$$

where $\mathbf{M}_i$ is the $i$-th predicted segmentation mask and $s_i$ is the confidence score. For each mask $\mathbf{M}_i$, we consider a binary label $y_i \in \{0, 1\}$ by computing its corresponding bounding box and using the IoU score, which assesses the quality of bounding boxes:

$$y_i = \begin{cases} 1 \text{ if } \max_j \text{IoU}(\mathbf{B}_i, \mathbf{O}_j) > \theta \\ 0 \text{ otherwise} \end{cases} \tag{6}$$

where $\theta > 0$ is a threshold, $\mathbf{B}_i$ is a bounding box computed from mask $\mathbf{M}_i$, and $\mathbf{O}_j$ is a mask that represents answer object $o_j$. We extracted instances from DeiVG datasets not used in VETO training (ca. 2000 samples), which are divided into training, validation, and test splits that contain 1200, 400, and 400 instances, respectively. If no targets are detected by the forward reasoner, DeiSAM produces a mask of a randomly chosen object in the scene, *i.e.*, if the maximum confidence score for targets is less than 0.2, DeiSAM was set to sample a random object in the scene and segment with a confidence score randomly sampled from a uniform distribution with a range of $[0.1, 0.4]$.

**Optimization.** We used the RMSProp optimizer with a learning rate of $1e - 2$, and performed 200 steps of weight updates with a batch size of 1. The reasoners' inference step was set to 4. We used IoU score's threshold $\theta = 0.8$.

## F Ablations

Subsequently, we present detailed results corresponding to the ablation studies in Sec. 5.4.

**LLMs for Rule Generation and Semantic Unification.** First, we evaluated multiple open and closed-source language models of different sizes on rule generation. The results on correct predicate identification and rule generation are reported in Tab. 7. In general, all instruction-tuned models can generate logic rules from deictic prompts. However, larger models strongly outperform smaller

---

[12] https://huggingface.co/xdecoder/SEEM/resolve/main/seem_focall_v0.pt

Table 7: Performance in logic rule generation of various large language models. We report the success rate of extracting the correct predicates and generating the correct rules in isolation since the predicates are supplied as context for rule generation. The combined success rate are the percentage of samples were both are correct.

| Model | Corr. Predicates (%) ↑ | | | Corr. Rules (%) ↑ | | | Combined (%) ↑ | | |
|---|---|---|---|---|---|---|---|---|---|
| | $DeiVG_1$ | $DeiVG_2$ | $DeiVG_3$ | $DeiVG_1$ | $DeiVG_2$ | $DeiVG_3$ | $DeiVG_1$ | $DeiVG_2$ | $DeiVG_3$ |
| Mistral-7B-Instruct | 75.50 | 75.52 | 67.80 | 86.95 | 82.56 | 75.43 | 65.88 | 64.63 | 51.50 |
| Llama-2-7B-Chat | 75.03 | 26.91 | 12.47 | **98.29** | 96.36 | 95.22 | 74.06 | 25.92 | 12.06 |
| Llama-2-13B-Chat | 92.87 | 97.19 | **96.05** | 97.88 | **97.82** | 97.13 | 91.00 | 95.17 | **93.45** |
| GPT-3.5-turbo | **96.57** | **97.43** | 93.21 | 97.45 | 96.96 | 95.04 | **95.66** | 95.36 | 89.92 |

Table 8: Comparison of various embedding models on the semantic unification step in the DeiSAM pipeline. The task is to identify one synonymous object name out of all objects in a scene, given their embedding similarity. Success rate is reported over ca. 2.5k scenes from Visual Genome.

| Model | Unification Success (%) ↑ |
|---|---|
| Glove-Wiki-Gigaword | 52.27 |
| Word2Vec-Google-News | 55.03 |
| OpenAI-CLIP ViT-B/32 | 52.90 |
| DFN5B-CLIP ViT-H/14 | 66.65 |
| MonarchMixer-Bert | 30.15 |
| MPNet-Base-v2 | **71.62** |
| MiniLM-L6-v2 | 62.79 |
| OpenAI-ada-002 | 66.50 |

ones, especially for more complex inputs. Interestingly, the types of failure cases differ significantly between models. For example, both Llama models and GPT-3.5-turbo rarely make syntactical errors in rule generation ($< 1\%$), whereas most of Mistral-7B's incorrect rules are already syntactically unsound.

For the semantic unification task we compare various semantic embedding models as shown in Table 8.

**Runtime Analysis.** We provide comparisons of the runtime of Grounded-SAM, LISA, and DeiSAM in Tab. 9. It shows the inference time per instance (a visual input with a textual prompt), averaged over 1000 examples for each dataset. GroundedSAM achieved remarkably faster inferences since it does not encode the reasoning process. Both LISA and DeiSAM approaches require about 6 to 10 seconds per example. For more analysis, we provide the running time for each component of DeiSAM in Tab. 10. It shows the inference time per instance of rule generation, semantic unification, differentiable forward reasoning, and segmentation. We observe that semantic unification is the most time-consuming process in the DeiSAM pipeline. In contrast, the differentiable forward reasoning and segmentation only make up a negligible fraction of the runtime. However, as outlined in the paper, the rule generation and semantic unification step rely on the OpenAI API and could be significantly reduced by running a local model instead.

Table 9: Runtime comparison of DeiSAM and baselines.

| | Running Time (sec) | |
|---|---|---|
| Method | $DeiVG_2$ | $DeiVG_3$ |
| GroundedSAM | $0.01 \pm 0.00$ | $0.01 \pm 0.00$ |
| LISA | $6.38 \pm 1.1$ | $7.03 \pm 1.17$ |
| DeiSAM (ours) | $9.98 \pm 4.61$ | $10.85 \pm 3.19$ |

Table 10: DeiSAM runtime analysis.

| | Running Time (sec) | |
|---|---|---|
| Method | $DeiVG_2$ | $DeiVG_3$ |
| Rule Generation | $1.4 \pm 0.63$ | $1.76 \pm 0.67$ |
| Semantic Unification | $6.38 \pm 1.1$ | $7.03 \pm 1.17$ |
| Forward Reasoning | $0.18 \pm 0.67$ | $0.18 \pm 0.31$ |
| Segmentation | $1.22 \times 10^{-6}$ | $9.82 \times 10^{-7}$ |

**DeiVG in the ReCOCO's reference format.** We conducted additional experiments by modifying the deictic prompts in DeiVG datasets to the ones with targets' labels, resulting in prompts similar to reference texts in RefCOCOs (Yu et al., 2016). Table 11 shows the mAP DeiVG datasets with the modified prompts using LISA, GroundedSAM, and DeiSAM. ± represents the gain compared to the original performance with deictic prompts (shown in Tab. 1). Neural baselines gained the

performance remarkably by the modified prompts since they contain labels for targets (*e.g.*person, kid), on which neural models highly rely to identify targets.

Table 11: Comparison with modified prompts, e.g. *"Person on the boat"* instead of "An object on the boat". $\pm$ represents the gain compared to the original performance with deictic prompts.

| Method | Mean Average Precision (%) $\uparrow$ | | |
|---|---|---|---|
| | DeiVG$_1$ | DeiVG$_2$ | DeiVG$_3$ |
| LISA | 26.81 (+11.91) | 68.78 (+12.75) | 82.20 (+6.41) |
| GroundedSAM | 21.87 (+11.39) | 47.91 (+15.58) | 89.48 (+43.44) |
| DeiSAM (ours) | 64.78 (−0.36) | 84.23 (−1.17) | 88.19 (+0.36) |

# G  Additional Segmentation Results

We provide supplementary results of the segmentation on the DeiVG$_2$ dataset in Fig. 12.

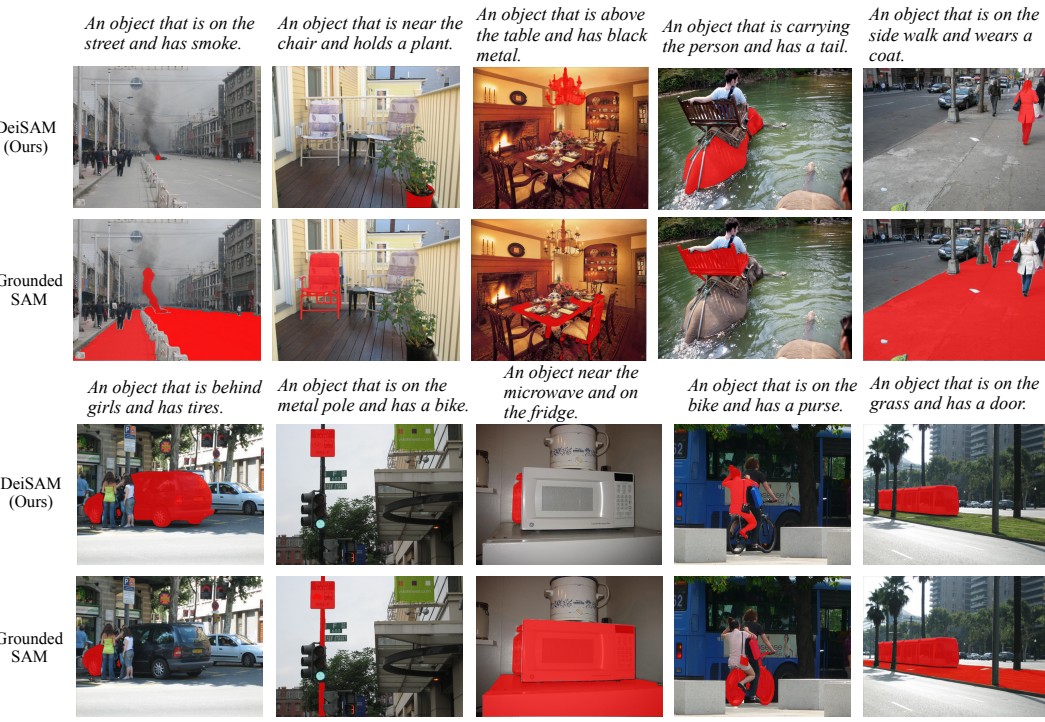

Figure 12: Segmentation results on the DeiVG$_2$ dataset using DeiSAM and GroundedSAM with deictic prompts (top).

