# OpenReview forum: "DeiSAM: Segment Anything with Deictic Prompting"
_NeurIPS.cc/2024/Conference — NeurIPS 2024 poster_

### Official Review · Reviewer_9P1c · 2024-06-16

**Soundness:** 2
**Presentation:** 3
**Contribution:** 2
**Rating:** 3
**Confidence:** 5

**Summary:**

This paper introduces a new reasoning-based segmentation by adding deictic logic as prompts into the SAM. Motivated by the deictic logical analysis, the authors present DeiSAM, a combination of large pre-trained neural networks with differentiable logic reasoners for deictic promptable segmentation. DeiSAM first use LLMs to generate first-order logic rules and perform differentiable forward reasoning on generated scene graphs. In addition, authors present Deictic Visual Genome dataset. Experimental results show DeiSAM works better than pure data-driven baselines.

**Strengths:**

1, Overall writing is good and easy to follow. Motivation is easy to understand.

2, The performance on proposed DeiVG benchmark looks good.

3, The idea of using first order deictic as prompting for segmentation is interesting.

**Weaknesses:**

Despite the idea of combining first order deictic prompt into SAM is interesting, there are still lot of limitations in this work.

1, Motivation is not new. Performing reasoning segmentation via LLM is not new. There several previous works exploring the same settings [1]-[5]. The difference lies on the author adopt more visual cues (scene graph) into LLM before sending the language features to SAM.

[1], LISA: Reasoning Segmentation via Large Language Model, CVPR-2024.

[2], An Improved Baseline for Reasoning Segmentation with Large Language Model, arxiv-2023.

[3], GRES: Generalized Referring Expression Segmentation, CVPR-2023.

[4], Towards Robust Referring Image Segmentation, TIP-2023.

[5], See, Say, and Segment: Teaching LMMs to Overcome False Premises, CVPR-2024.

2, The technical novelties are limited. The proposed method just combines previous differentiable logic reasoners with LISA-like segmenter (LLM + SAM) architecture. I find no extra insights or advantages for end-to-end reasoning learning. Moreover, the method needs extra scene graphs as inputs, which introduce more complexities to the pipeline.

3, Related works discussion. Missing lots of works on referring segmentation [3]-[5].

4, Performance and benchmarks. There are only refcoco+ and proposed benchmark results. Several important benchmarks are missing (Please refer to [1]). Moreover, the performance is not strong. This result is misleading since the authors miss lots of previous works for fair comparison.

5, Missing detailed ablation studies for each component. It is hard to know the real benefits on the combined logical reasoner and effects of input scene graph.

**Questions:**

See the weakness part.

**Limitations:**

Yes, they have checklist.

---

> ### Author Rebuttal · Authors · 2024-08-06
>
> We thank the reviewer for the fruitful feedback and for acknowledging that the paper is well-written and that combining first-order logic with prompting is interesting.
>
> We would like to address concerns next.
>
> > Motivation is not new. Performing reasoning segmentation via LLM is not new. There several previous works exploring the same settings [1]-[5].
>
> We disagree with this. Our claim is to use something other than LLMs for reasoning, as they often hallucinate on complex prompts. Instead, we propose using logic for reasoning. We integrate LLMs and SAM with differentiable reasoners for neuro-symbolic inference on various deictic prompts. To the best of our knowledge, this approach has not been attempted in previous studies.
>
> Most importantly, we demonstrated that LISA [1] significantly degrades its performance on abstract prompts in Tables 3 and 4, indicating a divergence from our problem setting. Our focus does not encompass referring expressions or commonsense reasoning, which differ from high-level abstract reasoning.
>
>
>
>
> > The difference lies on the author adopt more visual cues (scene graph) into LLM before sending the language features to SAM.
>
> Not quite. There is a factual error: we do not feed scene graphs to LLMs, nor do we utilize LLMs directly for reasoning. In DeiSAM, LLMs generate logic programs to reason with scene graphs on the abstract level.
>
>
> > The technical novelties are limited.  The proposed method just combines previous differentiable logic reasoners with LISA-like segmenter (LLM + SAM) architecture.
>
> As Reviewer YJtK pointed out, our major findings are as follows:
> - Existing strong baselines using transformers for reasoning perform poorly on the segmentation tasks with abstract complex prompts.
> - This indicates that the current transformer-baed models are limited in high-level abstract visual reasoning.
> - DeiSAM is the first framework to successfully extend segmentation models using differentiable reasoning with scene graphs to gain reasoning ability over abstract representations.
>
>
>
>  > I find no extra insights or advantages for end-to-end reasoning learning.
>
> We address the trade-off between logic-based reasoning and model adaptability: while logic provides faithful reasoning capabilities, it forces the model to be deterministic and unlearnable. For instance, employing off-the-shelf logic libraries such as Prolog ensures reasoning but impedes gradient-based learning due to their non-differentiable inference. The key insight of our approach is that it enables the segmentation model to function as both a faithful reasoner and an adaptive learner, as demonstrated in Table 5.
>
>
> >  Moreover, the method needs extra scene graphs as inputs, which introduce more complexities to the pipeline.
>
> Not quite. There is indeed a trade-off between the complexity of the input and the reasoning models employed. Scene-graph representations make the reasoning module highly parameter-efficient. For instance, in DeiSAM, the forward reasoner with $N$ rules requires only $N$ parameters, with one weight per rule.
> In contrast, purely neural models typically require a large number of parameters in the reasoning model (e.g., experiments in [5] employ LLaVA-v1.5-7B as the LLM backbone) to perform reasoning on pixels.
>
>
> >  4, Performance and benchmarks. There are only refcoco+ and proposed benchmark results. Several important benchmarks are missing (Please refer to [1]).
>
> In our manuscript, on lines 90-95, we explicitly discuss the reasons for not conducting experiments on the benchmark [1]. The suggested benchmark [1] primarily focuses on low-level commonsense reasoning, whereas our objective is to address high-level abstract reasoning segmentation. This rationale also extends to other referring-expression segmentation benchmarks.
>
>
> > Moreover, the performance is not strong. This result is misleading since the authors miss lots of previous works for fair comparison.
>
> We respectfully disagree with the suggestion that additional baselines are necessary to validate our experiments. We consistently used the LISA [1] model, a state-of-the-art segmentation method, as the baseline for our experiments. None of the works suggested by the reviewer specifically address high-level abstract reasoning. Therefore, we believe that incorporating these additional baselines is not essential to validate our experimental results.
>
> > Missing detailed ablation studies for each component. It is hard to know the real benefits on the combined logical reasoner and effects of input scene graph.
>
>
> We refer to the general remark. To answer this, we conducted additional experiments on abstract visual scenes with more complex prompts and demonstrated the gain by having the logical reasoner with scene graphs.
>
> > Missing lots of works on referring segmentation [3]-[5].
>
> We agree with the reviewer that papers [3]-[5] are relevant to our study. Here, we discuss these works and clarify our paper's contribution.
>
> GRES [3] proposes two distinct attention mechanism to address addresses multi-target segmentation. R-RIS [4] proposes  RefSegformer, a transformer-based model that includes a language encoder, a vision encoder, and an encoder-fusion meta-architecture  for handling incorrectly described textual prompts. See, Say, and Segment [5] propose SESAME, an LMM designed to "see" whether objects are present, "say" to interact with users, and "segment" target objects.
>
> All these methods rely on transformers (or attentions) as their core reasoning pipeline, and they would inherit the reasoning limitations inherent to purely neural models. In contrast, DeiSAM explicitly encodes logical reasoning processes to guarantee accurate and faithful interpretation of abstract and complex prompts.  We will include these discussions in the final version.
>
>
> Thank you once again for your valuable feedback. We hope our response adequately addresses your concerns, and we would be pleased to answer any further questions you may have.

---

> > ### Author Response · Authors · 2024-08-12
> >
> > Dear Reviewer,
> >
> > We hope we have answered all your questions and resolved the outstanding concerns. As the discussion phase is coming to an end, we would appreciate if the reviewer engages in a discussion so that we can answer any further concerns, if any.
> >
> > Regards,
> >
> > Authors

---

> ### Comment · Area_Chair_jtw8 · 2024-08-13
> **Reviewer 9P1c: please respond to the authors' rebuttal**
>
> Dear reviewer,
>
> thanks for your participation in the NeurIPS peer review process.  We are waiting for your response to the rebuttal.  You gave a reject rating (3), with 5 weaknesses.
>
> Is the response from authors satisfactory?
> - If yes, are you planning to increase your score?
> - If no, could you help the authors understand how they can improve their paper in future versions?
>
> Thanks,
> AC

---

### Official Review · Reviewer_TxaQ · 2024-07-11

**Soundness:** 4
**Presentation:** 4
**Contribution:** 3
**Rating:** 6
**Confidence:** 5

**Summary:**

This paper proposes DeiSAM to enhance the ability of current state-of-the-art referring segmentation frameworks with deictic prompting. DeiSAM is an innovative approach that combines large pre-trained neural networks with differentiable logic reasoners to perform image segmentation based on complex, deictic textual prompts. The method leverages Large Language Models (LLMs) to generate first-order logic rules from textual descriptions and uses differentiable reasoning on scene graphs to identify and segment objects in images accordingly. The paper also introduces the Deictic Visual Genome (DeiVG) dataset for evaluating deictic segmentation and demonstrates DeiSAM's superiority over neural baselines through empirical results.

**Strengths:**

In general, this paper identifies an important problem in the intersection between referring image segmentation and scene graph generation, and proposes a simple yet effective solution to resolve it. The contribution is comprehensive, from my perspective, including the following aspects:

1. **Novelty and Innovation**: The paper proposes a novel framework that integrates Large Language Models (LLM) with visual logic reasoning for deictic image segmentation, addressing a gap in the field where current methods struggle with complex prompts. The symbolic rules designed in the pipeline look relevant to the visual programming concept, which is novel and technically elegant.

2. **Empirical Validation and Efficacy**: The authors introduce a new dataset, DeiVG, to test the capabilities of DeiSAM, and provide extensive experimental results that validate the effectiveness of their approach, showcasing clear improvements over existing baselines. Concretely, for DeiVG1, DeiVG2 and DeiVG3 respectively, the proposed method outperforms the previous state-of-the-art LISA by 50.24%, 29.37% and 12.04% in terms of Mean Average Precision.

3. **Simplicity and End-to-end Differentiable**: DeiSAM is both simple and effective, which makes it easy to follow. DeiSAM's modular architecture and the use of differentiable reasoning allow for end-to-end training, which is a significant advantage for adapting to complex downstream tasks and improving segmentation quality.

**Weaknesses:**

1.
The clarity of the paper could be further improved. Despite careful reading and familiarity with the field, several aspects remain confusing:

- **Training Process**
  - **Unclear Tuning**: What components are tuned during the training stage?
  - **Revision Recommendation**: Annotate which modules are tuned during training.

- **Scene Graph Generator**
  - **Pre-training Limitations**: The Scene Graph Generator appears pre-trained on Visual Genome, which annotates only a limited set of objects (e.g., persons, boats, cars).
  - **Generalizability Concerns**:
    - Will the proposed method work on images containing objects not annotated in Visual Genome?
    - If not, the title "Segment Anything with Deictic Prompting" may be an overstatement.

- **Semantic Unifier and Forward Reasoning**
  - **Ambiguous Illustration**: The explanation of these components lacks clarity.
    - What are the inputs and outputs of the Semantic Unifier?
    - What constitutes the "background knowledge" $\mathcal{B}$ in forward reasoning?
    - What is learned in the forward reasoning process?

2.
The proposed method should also be evaluated on other referring image segmentation benchmarks, such as RefCOCO, RefCOCOg, and RefCOCO+. This evaluation would help determine whether the proposed method is effective in general referring image segmentation settings, or if the integration of scene graph information potentially hinders performance in standard referring segmentation tasks.

**Questions:**

All are listed in weaknesses. Please see above.

**Limitations:**

The authors have adequately addressed the limitations in the submitted manuscript.

---

> ### Author Rebuttal · Authors · 2024-08-06
>
> We thank the reviewer for the fruitful feedback and for acknowledging that the proposed framework is novel and effective and the empirical results are valid.
>
> We would like to address the concerns raised by the reviewer.
>
> > The proposed method should also be evaluated on other referring image segmentation benchmarks, such as RefCOCO, RefCOCOg, and RefCOCO+
>
> We already reported this in the paper. Table 3 shows the result on the RefCOCO+ benchmark. We observed that DeiSAM can handle the referring expression task, which is the common benchmark suggested by the reviewer.
>
> Moreover, in Table 4, we demonstrated that neural baselines (LISA and GroundedSAM) significantly degraded their performance if we modified the textual input to the abstract form, e.g., “kid wearing navy shirt” is modified to “an object that is wearing navy shirt.” This indicates the lack of the ability to use abstract high-level reasoning in neural baselines.
>
>
> > What components are tuned during the training stage? Annotate the tuned modules.
>
> The rule weights in the differentiable forward reasoner. On line 321, we explicitly stated: “We minimize the binary cross entropy loss with respect to rule weights $w_1$ and $w_2$.”
> To enhance clarity, we added annotations to the figure as the reviewer proposed. Thank you for the suggestion.
>
>
> > Pre-training Limitations: The Scene Graph Generator appears pre-trained on Visual Genome, which annotates only a limited set of objects (e.g., persons, boats, cars).
>
>
> We agree with the reviewer that good scene graph representations are the key to the proposed DeiSAM framework. Our finding is that if good scene graphs (generators) are available, the segmentation quality on abstract prompts can be much improved by combining logical reasoning.
>
> As scene graphs can be noisy and low-quality, DeiSAM performs embedding-based vocabulary matching and learning scene graphs in combination. It is a vital open question to handle scene understanding for unseen data in a zero-shot manner.
>
>
>
> > Will the proposed method work on images containing objects not annotated in Visual Genome?
>
> Yes, it works. We conducted additional experiments using the CLEVR environment [1] that contains abstract 3D visual scenes, which is largely different from visual genome. We show that DeiSAM can segment abstract 3D objects given more complex textual prompts.
> Moreover, our result on the RefCOCO+ dataset also validates that it is not limited to only visual genome images.
>
> [1] CLEVR: A Diagnostic Dataset for Compositional Language and Elementary Visual Reasoning. CVPR 2017
>
> > If not, the title "Segment Anything with Deictic Prompting" may be an overstatement.
>
> As the premise is false (we clarified above), we believe this is not an overstatement.
>
> > What are the inputs and outputs of the Semantic Unifier?
>
> The input is a scene graph and a set of rules generated by LLMs from textual prompts. If a term in the rules is absent in the scene graph, the semantic unifier investigates the most likely matching between these two representations using term embedding.
> We added these technical details to the final version of our manuscript Thank you for pointing this out.
>
>
> > What constitutes the "background knowledge"  in forward reasoning?
>
> It is a set of rules and facts described in first-order logic. It is common for logical reasoners to accept background knowledge from users. For DeiSAM, we provided no background knowledge throughout the experiments.
>
> > What is learned in the forward reasoning process?
>
> In our experiments, rule weights are learned for composing a mixture of scene graphs to maximize segmentation performance.
>
> To clarify, our forward reasoner implements forward-chaining reasoning in first-order logic, a method used to deduce all facts given known rules and known premises. This reasoning process is inherently deterministic. The differentiable version of the forward reasoner introduces weighted rules, which can be optimized through backpropagation. Our approach seamlessly integrates forward reasoning with neural modules, thereby enhancing the overall learning performance.
>
>
>
> Thank you once again for your valuable suggestions. We hope our response addresses your concerns, and we are happy to answer any further questions you may have.

---

> > ### Author Response · Authors · 2024-08-13
> >
> > Dear Reviewer,
> >
> > We hope we have answered all your questions and resolved the outstanding concerns. As the discussion phase is coming to an end, we would appreciate if the reviewer engages in a discussion so that we can answer any further concerns, if any.
> >
> > Regards,
> >
> > Authors

---

> ### Comment · Area_Chair_jtw8 · 2024-08-13
> **Reviewer TxaQ: please respond to the authors' rebuttal**
>
> Dear reviewer,
>
> thanks for your participation in the NeurIPS peer review process. You indicated that you are leaning towards accepting the paper. Is the response from authors satisfactory? Does it address weaknesses that you mentioned in the review?
>
> - If yes, are you planning to increase your score?
> - If no, could you help the authors understand how they can improve their paper in future versions?
>
> Thanks,
> AC

---

### Official Review · Reviewer_YJtK · 2024-07-12

**Soundness:** 3
**Presentation:** 4
**Contribution:** 2
**Rating:** 6
**Confidence:** 4

**Summary:**

The paper proposes to study Deictic references/prompts, i.e., phrases/references that describe the role/purpose/context rather than naming the object directly. Firstly, the paper constructs a new dataset with deictic prompts based on Visual Genome; Next, it is shown that existing methods do poorly on this task. Next, a neuro-symbolic method that works on top of a generated scene graph is proposed which is shown to work a lot better on this newly constructed dataset as well as perform comparably on related tasks such as referring expression recognition.

[EDIT: Post rebuttal comments are useful but do not change overall score]

**Strengths:**

**[S1] Interesting Findings:** While the setup is somewhat artificial/contrived (see W1), the results on existing methods are surprisingly poor for the proposed task. This is an interesting finding and might point to a bigger problem in the way "referring segmentation" is done currently. While there are existing datasets with referring expressions, I think the paper studies a more challenging setup which is harder to game so it should provide a clearer picture of true understanding capability of the models.

**[S2] Good clarity:** The presentation is clear and detailed. The experimental designs are meaningful and the results are clearly presented.

**Weaknesses:**

**[W1] Somewhat contrived setup:** I fully agree with the big premise that deictic expressions are important and common in everyday usage. However, many examples from the constructed dataset are unnatural and forced. The paper also takes deictic phrases to an extreme by removing all nouns from the deictic expressions with very generic prompts such as "stuff" or "object". It is more common for references to consist of categorial nouns instead of object names (E.g., that toy on top of the chair) instead of teddy bear, etc. So, while a useful tool in diagnostic abilities, I feel that the dataset is not broadly applicable. Similarly, since it is such an unnatural setup; It might also make it unrealistically hard for general purpose baseline methods.

**[W2] Reliance on Scene graphs:** The proposed work relies on access to scene graph prediction model outputs. This is not something that is available to baselines and, in my opinion, does a lot of heavy lifting. I believe this makes it hard to do a fair comparison with the baselines. One could even argue that if the scene graph is comprehensive and accurate the **main** task in decoding deictic expressions (Thought I admit that it is still non trivial). Also see Q1.

[W3] Minor issues on the dataset: The dataset is divided based on how many "hops" of reasoning are necessary to decode the expression. E.g., "A tall object wearing a hat walking in sidewalk" is a 3 hop expression (tall, wearing a hat, walking on the sidewalk). However, unless there are counterexamples to each. (E.g., other tall things, other stuff wearing the hat, ...) it degenerates into a subset of that. This means a shortcut "wearing a hat" alone might be enough to decode the object. I would like to see an analysis of this.

**Questions:**

Q1. If scene graphs are assumed to be available, should they not be available in some form to every algorithm. E.g., if the object name is easily decodable from deictic expression by looking it up in the scene graph, can we not change the expression to include the name of the object as well? E.g., "A red furry object" --> Select all objects that are red and furry from scene graph --> Grab the names of those objects -> use the name instead of deictic expression when prompting SEEM or LISA.

Q2. What is the effect of the quality of scene graphs on the final results of the proposed method. Is it resitant to mistakes in the SGG prediction?

---

> ### Author Rebuttal · Authors · 2024-08-06
>
> We thank the reviewer for the detailed and fruitful feedback and for acknowledging that our findings are insightful, the experimental setups are meaningful, and the paper is well-written.
>
> We address the concerns next.
>
> > W1: many examples from the constructed dataset are unnatural by removing all nouns from the deictic expressions with "stuff" or "object".
> >  It is more common for references to consist of categorial nouns instead of object names (E.g., that toy on top of the chair) instead of teddy bear
>
> We agree with this. Using categorical names (e.g. toy, food, deice) would be very promising in covering more real-world applications of referring expressions. This could be achieved by having an LLM while dataset generation to convert each specific name to a categorical name.
>
> However, we do not conduct experiments on this as our primary focus is abstract reasoning in which object and categorical names are absent in the prompt, i.e. the models need to deduce the object name from a given relational specification.
>
> To mitigate, we reported on Table 10 in the Appendix of performances by making DeiVG datasets with a more referring-expression style, e.g. we provide "Person on the boat" instead of “An object on the boat”. The results indicate that neural baselines gain a lot by having object names. We expect that if we provide the categorical names in the prompt, neural baselines will gain some performance but not as much as with object names.
>
>
> > W2:   Reliance on Scene graphs: The proposed work relies on access to scene graph prediction model outputs.
>
> We answer this later for Q1.
>
> > W3: Shortcut of in the DeiVG dataset. E.g.  "wearing a hat" alone might be enough to decode the object given prompt: tall, wearing a hat, walking on the sidewalk.
>
> We agree with the reviewer that this is often the case in our dataset. We have analyzed how often it happens in the generated DeiVG dataset.
>
> For example, with a prompt “An object wearing a hat walking on the sidewalk", we consider the following variations of ways to match objects:
>
> - Relation: We consider only relations to detect matching between objects in the scene, i.e., "wearing" or "walking on," disregarding the object names.
> - Object: We consider only objects to detect matching between objects in the scene, i,e.,  "hat" or "sidewalk," disregarding the relations.
> - Either: We consider either a partial match from Relation or Object.
> - Complete: We consider complete counterexamples, i.e., "wearing hat" or "walking on the sidewalk", which is exactly articulated by the reviewer.
>
> The table below shows the proportion of prompts involved with such shortcuts.
>
> | Proportion (%) | Relation | Object | Either | Complete |
> |----------------|------------------|----------------|----------------|-------------------------|
> | DeiVG$_3$        | 23.01            | 46.00          | 17.35          | 60.56                   |
> | DeiVG$_2$        | 23.39            | 48.22          | 18.13          | 58.65                   |
>
> The result indicates that the shortcut often happens in the dataset. In DeiVG$_3$, 60\% of the prompts are involved with such a shortcut, i.e., no other objects in the scene share the same 3 relations. We included this result to the appendix of our manuscript. Thank you for the suggestion to improve the paper.
>
>
> Moreover, to explore more cases with similar objects without shortcuts, we conducted an additional experiment on an abstract 3D environment with more complex prompts. For more details, refer to the general remark.
>
> > Q1: Can we use scene graphs for other baselines? E.g., "A red furry object" --> Select all objects that are red and furry from scene graph --> Grab the names of those objects -> use the name instead of deictic expression when prompting SEEM or LISA.
>
> This approach would not be fair. While it is an interesting suggestion to use scene graphs for neural baselines, the proposed method involves providing an external program for preprocessing the scene graphs so that neural models can process them. This would result in a composite model consisting of a scene graph preprocessor and a separate large segmentation model. Such an approach severely limits further extensions aimed at enhancing scene understanding and reasoning segmentation because it is not straightforward to propagate segmentation errors back to improve scene graph generation.
>
> In contrast, DeiSAM treats scene graphs as an internal primal representation. It performs segmentation through differentiable reasoning, allowing it to backpropagate errors to the scene graphs.  This capability is crucial for enabling the model to learn to segment more accurately from data and to generate explanations using gradients, indicating which parts of the scene were essential for the segmentation.
>
> An alternative comparison could be to encode the scene graphs as text and provide them to the neural baselines.
>
>
> > Q2: What is the effect of the quality of scene graphs on the final results of the proposed method. Is it resitant to mistakes in the SGG prediction?
>
> It significantly impacts the segmentation quality, but DeiSAM demonstrates resilience to noise in SGGs. If the quality of scene graphs is extremely low, the resulting segmentation quality will also be low, as the reasoning process relies on the scene graph representation. However, DeiSAM manages noise in scene graphs through embedding-based vocabulary matching and by learning a mixture of scene graphs.
>
>
> Thank you for your valuable suggestions. We hope our response satisfactorily addresses your concerns, and we are happy to answer any additional questions you may have.

---

> > ### Author Response · Authors · 2024-08-13
> >
> > Dear Reviewer,
> >
> > We hope we have answered all your questions and resolved the outstanding concerns. As the discussion phase is coming to an end, we would appreciate if the reviewer engages in a discussion so that we can answer any further concerns, if any.
> >
> > Regards,
> >
> > Authors

---

> > ### Comment · Reviewer_YJtK · 2024-08-13
> > **Post-rebuttal comments**
> >
> > I have read the responses to my comments as well as all other reviews and their responses carefully. Like me, others also had concerns about the use of scene graphs and limited vocals caused by using a COCO-bases scene graph. There hasn't been a very satisfactory response about this fact.
> >
> > Secon point that I raised was about "unfair" use of extra information in form of scene graph. Unfortunately, that also hasn't been very satisfactory.
> >
> > > This approach would not be fair. While it is an interesting suggestion to use scene graphs for neural baselines, the proposed method involves providing an external program for preprocessing the scene graphs so that neural models can process them. This would result in a composite model consisting of a scene graph preprocessor and a separate large segmentation model.
> >
> > I disagree. Scene graph representation in form of scene graph triplets can be natively processed by LLMs and even better with a few in-context examples. Even if needed to make a "program" it can be a very simple one based on semantic parsers (which can be inaccurate but would still provide a sort of a equalization of available information.
> >
> > > Such an approach severely limits further extensions aimed at enhancing scene understanding and reasoning segmentation because it is not straightforward to propagate segmentation errors back to improve scene graph generation.
> >
> > Yes it would, but that is fine and would only serve to differentiate the submitted work with the above approach, which is a strength. We would still have liked to see what the results are compared to a hypothetical naive model that gets the equivalent amount of information.
> >
> > However, both those points were originally included in my first round review so the scores remain unchanged.

---

> > > ### Author Response · Authors · 2024-08-14
> > > **Thank you for your response**
> > >
> > > Thank you for your response. We agree with the reviewer that the experimental setup suggested in the response for the LLM baselines would work as another naive baseline, on top of the LLM baselines presented in the paper. We will run the experiments with the suggested setup and include the results to the final version of our manuscript. Thank you again for your insightful suggestions to improve the paper.

---

> ### Comment · Area_Chair_jtw8 · 2024-08-13
> **Reviewer YJtK: please respond to the authors' rebuttal**
>
> Dear reviewer,
>
> thanks for your participation in the NeurIPS peer review process. You indicated that you are leaning towards accepting the paper. Is the response from authors satisfactory? Does it address weaknesses that you mentioned in the review?
>
> - If yes, are you planning to increase your score?
> - If no, could you help the authors understand how they can improve their paper in future versions?
>
> Thanks,
> AC

---

### Official Review · Reviewer_WXVv · 2024-07-13

**Soundness:** 3
**Presentation:** 3
**Contribution:** 3
**Rating:** 4
**Confidence:** 4

**Summary:**

The manuscript proposes DeiSAM, a framework that integrates large pre-trained neural networks with differentiable logic reasoners to address deictic promptable segmentation.
DeiSAM leverages large language models (LLMs) to generate first-order logic rules and performs forward reasoning on generated scene graphs for object segmentation based on complex textual prompts.
The Deictic Visual Genome (DeiVG) dataset is introduced for evaluation.

**Strengths:**

1. The paper is well-structured and easy to read.
2. The integration of LLMs with differentiable logic reasoners for deictic prompt-based segmentation is a novel approach, addressing limitations in neural network-based segmentation models.

**Weaknesses:**

1. Over-simplified Assumptions: The curated datasets DeiVG1, DeiVG2, and DeiVG3 contain prompts using only 1-3 relations and are restricted to avoid multiple objects with ambiguity. However, in real-world applications, object and relation categories can be varied and rich, with inevitable visual-language alignment gaps. For example, in an image with a black cat and a white cat, how would the method handle a prompt to segment the white cat using relative descriptions? Moreover, the dataset size is small (10k), making large models prone to overfitting.

2. Dependence on Scene Graph Quality: The method heavily relies on the quality of scene graphs. Inaccurate scene graphs can lead to incorrect segmentation results, which is a critical limitation for practical applications. For instance, the large performance gap indicated in Table 5 highlights this dependency.

3. Unclear Philosophy: The use of a semantic unifier (an LLM) to associate objects in the prompt and scene graph seems trivial. The rationale for using a differentiable forward reasoner instead of continuing to use an LLM is not well-justified. The necessity and advantage of having a "differentiable" reasoner are unclear. How does this reasoner fundamentally differ from previous neural-based methods like Grounding DINO or LISA?

4. Insufficient Experiments: The experiments are limited to relatively easy cases for prompt-based segmentation on a small-scale image dataset. The paper lacks evaluations on more challenging scenarios, such as multiple objects of the same category or prompts with more complicated relations. Comparisons with previous methods under these conditions would provide a more comprehensive evaluation of DeiSAM’s effectiveness.

5. Unclear Contributions: Modules of DeiSAM are borrowed from other existing works, and there is no novel architecture designed for the prompt-based segmentation. It appears more like a technique report for an engineering project instead of being a research paper suitable for a top-tier conference like NeurIPS.

**Questions:**

Detailed questions are referred to as Weaknesses.

**Limitations:**

The authors have discussed the limitations of the work.

---

> ### Author Rebuttal · Authors · 2024-08-06
>
> We thank the reviewer for the thoughtful comment and for acknowledging that the paper addresses limitations in neural segmentation models and is well-structured and easy to read.
> We address the concerns next.
>
> > Over-simplified Assumptions: How would the method handle a prompt to segment the white cat using relative descriptions?
>
> Our method can handle such cases. Let us describe how to do that.
> Given prompt: “Segment a white cat”, an LLM can generate rules in our format:
> ```
> cond1(X):-type(X,Y),type(Y,white_cat).
> target(X):-cond1(X).
> ```
> Then, the semantic unifier will consider the embedding of the term `white_cat` and match it to an entity in the scene graph representing a white cat. The embedding-based semantic unification can address the visual-language alignment. If the scene graph misses entities of a white cat and a black cat in the first place, it would be hard for DeiSAM to perform reasoning correctly. A crucial unresolved issue is how to generate scene graphs in a zero-shot manner, akin to large visual-language models.
>
>
> > Moreover, the dataset size is small (10k), making large models prone to overfitting.
>
> The DeiVG dataset is proposed to evaluate segmentation models, and thus we believe that the dataset with 10k examples is not extremely small for this usage.
>
> >  Inaccurate scene graphs can lead to incorrect segmentation results, which is a critical limitation for practical applications.
>
> We argue that segmentation models must understand the scenes to be faithful reasoners. Without this assumption, models can learn to *shortcut*, i.e., learn to pretend to answer via an incorrect scene understanding and reasoning [1]. This does not essentially solve the complex reasoning prompts that we aim to address in the paper.
>
> [1] Not All Neuro-Symbolic Concepts Are Created Equal: Analysis and Mitigation of Reasoning Shortcuts, NeurIPS 2023
>
>
> > the large performance gap indicated in Table 5 highlights this dependency.
>
> We agree with this. The quality of scene graphs significantly impacts the segmentation quality, but DeiSAM demonstrates resilience to noise in scene graphs. If the quality of scene graphs is extremely low, the resulting segmentation quality will also be low, as the reasoning process relies on the scene graph representation. However, DeiSAM manages noise in scene graphs through embedding-based vocabulary matching and by learning a mixture of scene graphs, as demonstrated in Table 5.
>
>
> > Unclear Philosophy:  The rationale for using a differentiable forward reasoner instead of continuing to use an LLM is not well-justified.
>
> The primary advantage is that segmentation models transform into reliable reasoners. LLMs often hallucinate on complex abstract prompts and scenes. To demonstrate this, we conducted additional experiments on 3D visual scenes with more complex prompts. For more details, refer to the general remark.
>
> > The necessity and advantage of having a "differentiable" reasoner are unclear.  How does this reasoner fundamentally differ from previous neural-based methods like Grounding DINO or LISA?
>
> The advantage of the resulting model lies in its ability to learn through gradients, thereby enhancing its performance. In contrast, using an off-the-shelf discrete logic system (e.g., Prolog) for a segmentation model would render the entire system static by blocking the gradient flow. This static nature would severely limit its applicability, as it would necessitate expert intervention to rewrite rules whenever they are imperfect. To reconcile explicit logical reasoning with gradient-based learning, we employ a differentiable logic reasoner. As illustrated in Table 5, DeiSAM obviates the need for expert efforts to improve segmentation performance.
>
>
> > The paper lacks evaluations on more challenging scenarios, such as multiple objects of the same category or prompts with more complicated relations.
>
> Refer to the general remark.
>
> > There is no novel architecture designed for the prompt-based segmentation. It appears more like a technique report for an engineering project instead of being a research paper suitable for a top-tier conference like NeurIPS.
>
> We disagree with the premise. DeiSAM addresses segmentation tasks involving abstract, complex prompts by integrating logic reasoners. Experimental results substantiate its effectiveness, demonstrating that it outperforms strong neural baselines.
>
> We believe that utilizing existing modules and a well-defined computational architecture should not be viewed as a disadvantage. We contend that the criteria for acceptance at a conference should be based on the overall contribution of the paper rather than solely on the introduction of a "novel architecture."
>
>
> Thank you for your insightful suggestions. We hope our response effectively addresses your concerns, and we are happy to answer any further questions you may have.

---

> > ### Author Response · Authors · 2024-08-12
> >
> > Dear Reviewer,
> >
> > We hope we have answered all your questions and resolved the outstanding concerns. As the discussion phase is coming to an end, we would appreciate if the reviewer engages in a discussion so that we can answer any further concerns, if any.
> >
> > Regards,
> >
> > Authors

---

> ### Comment · Area_Chair_jtw8 · 2024-08-13
> **Reviewer WXVv: please respond to the authors' rebuttal**
>
> Dear reviewer,
>
> thanks for your participation in the NeurIPS peer review process.  We are waiting for your response to the rebuttal.  You gave a borderline rating (4). Is the response from authors satisfactory? Does it address weaknesses that you mentioned in the review?
>
> - If yes, are you planning to increase your score?
> - If no, could you help the authors understand how they can improve their paper in future versions?
>
> Thanks,
> AC

---

### Author Rebuttal · Authors · 2024-08-06

We thank the reviewers for their thoughtful feedback and insights on the paper. Here, we would like to address concerns shared by several reviewers.

Reviewer WXVv and Reviewer 9P1c:
> What is the benefit of integrating (differentiable) logical reasoners into segmentation models?

The primary advantage is that segmentation models transform into reliable reasoners. It is well-documented that large language models (LLMs) often underperform on complex reasoning tasks [1,2]. Consequently, depending on LLMs within a segmentation model introduces an inherent bottleneck for abstract reasoning capabilities. This paper aims to propose a segmentation model capable of addressing abstract reasoning. Crucially, this approach is distinct from solving referential tasks, where abstract representations and complex reasoning do not play a central role.

[1] Large language models can be easily distracted by irrelevant context. ICML 2023

[2] Large language models cannot self-correct reasoning yet. ICLR 2024


This argument responds to a comment from Reviewer WXVv:
>  The paper lacks evaluations on more challenging scenarios, such as multiple objects of the same category or prompts with more complicated relations.

Reviewer 9P1c:

>  It is hard to know the real benefits on the combined logical reasoner and scene graphs.

To answer this,  we conducted additional experiments using the CLEVR environment, wherein multiple abstract objects (e.g., a large cube and a small sphere) are presented in 3D scenes. Our objective is to demonstrate that DeiSAM can handle such abstract objects with complex prompts while neural baselines frequently struggle to reason with them.


**Task.** The  task is to segment objects given prompts where the answers are derived by the reasoning over abstract list operations. We consider 2 operations: delete and sort. The input is a pair of an image and a prompt, e.g. “Segment the second left-most object after deleting a gray object?”.  Examples are shown in Fig. 1 in the attached PDF.  To solve this task, models need to understand the visual scenes and perform high-level abstract reasoning to segment.

**Dataset.**  **(Image)** We generated visual scenes using the CLEVR environment [1]. Each visual scene contains at most 3 objects with different attributes: (i) colors of cyan, gray, red, and yellow, (ii) shapes of sphere, cube, and cylinder, (iii) materials of metal and matte. We excluded color duplications in a single image. **(Prompts)** We generated prompts using a templates: “The [Position] object after [Operation]?”, where [Position] can take either of: left-most first, second, or third. [Operation] can take either of: (I) delete an object, and (II) sort the objects in the order of: cyan < gray < red < yellow (alphabetical order). We generated 10k examples for each operation.

**Models.** We employed DeiSAM with pre-trained Slot Attention [2] to perceive objects. DeiSAM initially learns abstract list operations from the visual Inductive Logic Programming (ILP) dataset [3], which includes positive and negative 3D visual scenes for each list operation, thereby deriving rules to perform these operations. These rules are then applied to segment objects, augmented by those generated from textual prompts. We used GroundedSAM and LISA for neural baselines. Hyperparameters are described in Section E in the Appendix.


**Results.**
The table below shows mAP for each baseline. The result indicates that segmentation models relying on neural reasoning pipelines fail to deduce segmentations for abstract reasoning prompts, while DeiSAM effectively identifies and segments the object specified by the prompt.

| mAP ( $\uparrow$) | Delete | Sort |
|---------------|---------|--------|
| DeiSAM | 99.29 | 99.57 |
| GroundedSAM | 7.6 | 15.39 |
| LISA | 12.88 | 11.15 |

Moreover, Fig. 1 in the attached PDF shows qualitative results. DeiSAM successfully segments objects with high-level reasoning, but neural baselines often failed to identify the correct target object.
We will add these results to the final version of the paper.

Most importantly, our primary goal is not to outperform neural baselines across all types of reasoning tasks. Instead, our aim is to demonstrate that neural baselines are insufficient for solving abstract reasoning prompts and to enhance their capabilities by integrating differentiable logic reasoners.

[1] CLEVR: A Diagnostic Dataset for Compositional Language and Elementary Visual Reasoning. CVPR 2017

[2] Object-Centric Learning with Slot Attention. NeurIPS 2020

[3] Learning Differentiable Logic Programs for Abstract Visual Reasoning. Mach. Learn., 2024

---

### Decision · Program_Chairs · 2024-09-25

**Decision:**

Accept (poster)

**Comment:**

Summary of Review Process:
- 4 reviews.
- Mixed scores 3, 4, 6, 6
- Authors submitted rebuttals/responses
- 3 out of 4 reviewers did not engage in discussion after repeated requests from AC. Only Reviewer YJtK (score 6) engaged in discussion.
- The AC has decided that the lack of participation from 3 reviewers should not negatively impact the final recommendation. Therefore the AC will review the questions and responses instead of directly considering the *scores* of the unresponsive reviewers.

Meta Review:
There was consensus from all reviewers regarding the novelty of the approach, clarity of writing, and experimental findings of the paper.  The following is a summary of questions and rebuttal:
- Reviewer 9P1c commented that the motivation was not new and the method used a combination of previous techniques. The AC believes that the response from authors sufficiently addresses this concern.
- Reviewer 9P1c suggested previous articles to be mentioned in related work. The authors have agreed to do so.
- Reviewer 9P1c asked for additional ablation studies. The authors have added results in the general remark. The AC recommends the authors to add these to the main paper.
- Reviewer TxaQ asked the authors to elaborate on some confusing aspects related to the technical approach, illustrations, and some limitations. The authors have responded sufficiently with elaboration/clarification and additional experiments.
- Reviewer WXVv asked the authors to comment on the assumptions and dependence on scene graph, provide additional experiments, and elaborate on the rationale and contributions. The authors have responded to each question. The AC recommends the authors to integrate these responses in the main paper.
- Reviewer YJtK participated in the discussion and gave a rating of Weak Accept. The reviewer states that some of the points still remain unaddressed.

Overall, the AC's evaluation is that the paper is an interesting work on neuro-symbolic approaches using LLMs to generate logical rules and perform differential reasoning on scene graphs.